# Evolution of the Microrobots: Stimuli-Responsive Materials and Additive Manufacturing Technologies Turn Small Structures into Microscale Robots

**DOI:** 10.3390/mi15020275

**Published:** 2024-02-15

**Authors:** Frank Marco den Hoed, Marco Carlotti, Stefano Palagi, Patrizio Raffa, Virgilio Mattoli

**Affiliations:** 1Center for Materials Interfaces, Istituto Italiano di Tecnologia, Via R. Piaggio 34, 56025 Pontedera, Italy; virgilio.mattoli@iit.it; 2Smart and Sustainable Polymeric Products, Engineering and Technology Institute Groningen (ENTEG), University of Groningen, Nijenborgh 4, 9747 AG Groningen, The Netherlands; p.raffa@rug.nl; 3Dipartimento di Chimica e Chimica Industriale, University of Pisa, Via Moruzzi 13, 56124 Pisa, Italy; 4BioRobotics Institute, Sant’Anna School of Advanced Studies, P.zza Martiri della Libertà 33, 56127 Pisa, Italy; stefano.palagi@santannapisa.it

**Keywords:** microrobotics, direct laser writing, microfabrication, microactuators, functional materials, untethered microdevices, microdevice integration, nanorobots

## Abstract

The development of functional microsystems and microrobots that have characterized the last decade is the result of a synergistic and effective interaction between the progress of fabrication techniques and the increased availability of smart and responsive materials to be employed in the latter. Functional structures on the microscale have been relevant for a vast plethora of technologies that find application in different sectors including automotive, sensing devices, and consumer electronics, but are now also entering medical clinics. Working on or inside the human body requires increasing complexity and functionality on an ever-smaller scale, which is becoming possible as a result of emerging technology and smart materials over the past decades. In recent years, additive manufacturing has risen to the forefront of this evolution as the most prominent method to fabricate complex 3D structures. In this review, we discuss the rapid 3D manufacturing techniques that have emerged and how they have enabled a great leap in microrobotic applications. The arrival of smart materials with inherent functionalities has propelled microrobots to great complexity and complex applications. We focus on which materials are important for actuation and what the possibilities are for supplying the required energy. Furthermore, we provide an updated view of a new generation of microrobots in terms of both materials and fabrication technology. While two-photon lithography may be the state-of-the-art technology at the moment, in terms of resolution and design freedom, new methods such as two-step are on the horizon. In the more distant future, innovations like molecular motors could make microscale robots redundant and bring about nanofabrication.

## 1. Introduction

In the last forty years, the idea of making robots so tiny that they can operate inside the human body has moved from the realm of science fiction to that of science [1]. Recent decades have seen exponential advances in the field, and the first microrobots (i.e., robots at the sub-millimeter scale) are now entering clinics [2]. Despite recent advances, microrobotics still present key open challenges, which concern the implementation of the very functionalities that characterize a robot: power, actuation, control, sensing, control, and intelligence [3]. Researchers have found ingenious solutions to achieve highly functional microrobotic systems while outsourcing most or all these functionalities to external driving subsystems. Recently, however, advances in smart polymeric materials and microfabrication technologies are enabling the development of intrinsically functional microrobots (with onboard actuation, sensing, etc.) [4]. A major contributor to this progress has been the emergence of 3D printing, which allows for unmatched versatility in the geometric freedom of microstructures and the use of advanced polymeric materials. Here, we review the emergence of novel responsive and active materials as well as microfabrication 3D additive manufacturing technologies and how they enabled key advances in the field of microrobotics (Figure 1).

A microrobot typically consists of a microfabricated structure that is powered and controlled by an external, larger system. The realization of microrobots that are mobile, i.e., untethered, and must thus be powered and controlled in a wireless manner poses even more challenges and design limitations. Usually, these devices are powered by external fields, such as magnetic, optical, electric, or acoustic fields, acting on the microrobot or some of its components. In this way, controlled forces and torques can be directly exerted on the microrobot body, achieving controlled motion through what can be called wireless actuation [5]. With this scheme, power, control, and actuation functionalities are not on board the microrobot but rather they are all taken up by the externally generated driving field. Often, there is no onboard sensing either, and the microrobot’s movement is tracked and controlled by image-based, eye-to-hand (or better, eye-to-microrobot) visual survey (i.e., by a camera fixed in the workspace). In other words, microrobots are often just small structures, while the ‘real robotics’ lie outside of the microrobot in a normal-scale system. This approach presents clear advantages in the design and fabrication of microrobots, but also strong limitations on their functionality. As novel advanced microfabrication techniques and smart responsive and active materials emerge, bringing these functionalities on board becomes possible, eventually enabling a new generation of advanced microrobots.

One major bottleneck in achieving self-contained microrobots is on-board actuation. Actuators are what allow robots to act on the environment, permitting their articulated movements and locomotion. Building actuators at the microscale is possible, either by MEMS approaches or by exploiting materials that respond to external stimuli with mechanical deformations. Analogously to actuators for normal-scale robots, traditional MEMS- or material-based micro-actuators are powered and controlled electrically. Whereas electrically powered micro-actuators (such as comb-drive actuators, shape-memory alloys, piezoelectric materials, and electro-active polymers) are a suitable choice for stationary microrobots, they are not suited to microrobots that are mobile and (consequently) untethered. These latter require actuators that can be powered and controlled in a wireless manner, such as, for instance, photoresponsive or magnetic materials. Moreover, actuators for mobile microrobots are often intended for self-propulsion of the microrobot and must thus be able to deform substantially (tens %) and/or at significant rates (tens to hundreds %/s) to achieve meaningful locomotion speeds. These strong requirements make the implementation of onboard actuation on mobile microrobots particularly challenging and fascinating. Within mobile microrobots, we can distinguish two types of operation. The first is defined as proximity operating, meaning that the driving field is generated at a distance, d, much smaller than the body length, *L*, (*d << L*), which includes microrobots driven by electric fields. In the second, which comprises remotely operating microrobots, the driving field is generated at a distance much larger than the body length (*d >> L*), such as in the case of microrobots driven by magnetic fields.

Here we provide a perspective on the current and emerging smart polymeric materials and 3D additive manufacturing technologies for implementing actuation in microscale robots. In doing so, we will not include silicon and other passive inorganic materials (which belong more to the realm of MEMS), and just briefly discuss biohybrid actuators, i.e., those that make use of cells or bacteria to actuate or propel microsystems (which were reviewed in detail recently; see Ref. [5]), as our focus is mainly on artificial functional systems.

**Figure 1 micromachines-15-00275-f001:**
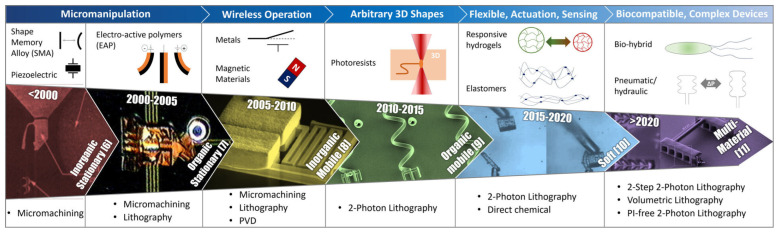
Schematic overview of the evolution of microscale robotic research since the beginning of the century. The scheme contains images that are characteristic of the period in which they were reported [6,7,8,9,10,11]. All images are used with permission from the original authors. Each microrobotic era opened the possibilities for new functionalities (blue bar), which were driven by innovation in both material and fabrication technologies.

## 2. Current Additive Manufacturing Fabrication Technologies for Microrobotics

An increasing number of techniques have become available for the fabrication of microrobotic devices. Naturally, there is a range of technologies beyond the scope of Additive Manufacturing including but not limited to glancing angle deposition, photolithography, electrodeposition, self-scrolling, chemical synthesis, self-assembly, and template-assisted (biological and electrochemical deposition) methods. All of these are valuable technologies for the structures but, nevertheless, cannot parallel the versatility and geometric freedom of 3-dimensional fabrication. These are crucial elements to achieve locomotion or other complex tasks at the microscale. Furthermore, 3D printing at the microscale is a relatively new method and its development is still accelerating as evidenced in Section 3. Additive manufacturing in microrobot fabrication technologies is either newly developed or improved versions of older techniques, which made remarkable progress in resolution, bringing them within the realm of microfabrication. Several other reviews have provided excellent descriptions of the various techniques suitable for fabricating microrobotics [12,13,14] or have given an overview of the historic developments of stereolithography [15]. We intend to provide an updated overview of the main materials and applications connected to 3D rapid manufacturing at the microscale as well as a peek into the future. Various very recent developments may well change the prospect of the field by increasingly widening applicability and accessibility. Table 1 shows an overview of the methods discussed combined with some key properties. Figure 2 provides a graphic overview of two of these key properties, resolution and printing speed, which give an idea of the range of applicability for each technique. The next paragraphs discuss all relevant fabrication technologies including several emerging ones and their microrobotic applications. The methods are also displayed schematically in Figure 3.

### 2.1. Stereolithography (SLA)

SLA is the most conventional laser-based microfabrication technique. It relies on a UV laser aimed at a tank containing a monomeric liquid and a stage. The laser initiates a cross-link polymerization reaction and thereby a solidified pattern in the liquid while motorized vertical movement of the stage enables layer-by-layer fabrication of 3-dimensional structures. Non-exposed material is washed away in a post-print development step.

The initial use of SLA in microrobot fabrication concerned microfluidic [34] and piezoelectric devices [35]. However, the limited resolution makes the technique unattractive for direct use in microrobotic fabrication. Rather, it became a widespread method to build molds and scaffolds for microstructures [15]. In addition, the wide range of available biodegradable resins makes it attractive for biomedical applications. 

### 2.2. Digital Light Processing (DLP)

The concept of digital light processing has a very long history starting in the 1960s, but only reached sufficient resolution for microfabrication around the beginning of this century [16]. DLP is similar in many respects to SLA but selectively projects the UV light onto an entire layer of photoresin instead of scanning singular points. This not only renders printing significantly faster but also allows for more reliable incorporation of nanoparticles into the photoresin, which can, for instance, be exploited by incorporating carbon nanotubes to build conductive paths [36]. The main microdevices that have been fabricated with DLP are in the field of microfluidics. Limitations in resolution have held back the development of more complex functional devices, but recent efforts to improve laser focus have resulted in project micro stereolithography, a subcategory in the field [37]. The first microactuator obtained with this technology was reported by Han et al. [38]; however, its relatively large size (several hundreds of micrometers) makes the fabrication approach unsuitable for devices of small dimensions. This example shows how, despite its tremendous achievements, DLP today is not among the preferred techniques when high resolution and precision at the micrometer scale are required. The development of large-area micro printers (LAMP) might change that in the future [17]. A further paramount challenge to address is the difficulty of combining multiple materials in a single structure, which is time-consuming and inaccurate [16].

### 2.3. Continuous Liquid Interface Production (CLIP)

Besides the pursuit for higher resolution, new methods seek to increase fabrication speed. In 2015, CLIP was reported for the first time, which finds speed by eliminating the recoating step required between each layer in DLP and SLA [18]. Instead, an oxygen-permeable window on the side of the light source of the resist tank provides a constant oxygen influx inhibiting the free radical reaction and creating a so-called ‘dead zone’. On the stage side of the tank, the oxygen concentration is sufficiently low for the reaction to proceed. For this reason, the vertical movement can proceed continuously, increasing fabrication speed or smoothness by increasing the number of programmed layers. The improved speed unfortunately comes with a price as the accuracy is reduced compared to conventional DLP. A more fundamental issue that comes with continuous printing is the necessity to use low-viscosity photoresists, dramatically limiting the available materials. Possibly, it is for this reason that no microrobotic devices fabricated by CLIP have been reported.

### 2.4. Two-Photon Lithography (TPL)

Also called direct laser writing (DLW), TPL uses an 800 nm femtosecond laser focused on a UV-absorbing photopolymerizable resist. In the focal point, the intensity of the radiation is high enough to trigger non-linear absorption phenomena such as two-photon absorption, e.g., a single UV-absorbing photoinitiator can absorb two (or more) NIR (near infrared) photons and induce cross-linking in the resist (usually comprising poly-functional acrylates). As this only happens at the focal point, it is possible to fabricate highly accurate micro-structures with submicrometric resolution by moving the focal point in three dimensions with galvo mirrors and piezoelectric motors [39]. Over the past decade, TPL has manifested itself as a state-of-the-art high-resolution fabrication method for microfabrication. The combination of high resolution (feature size < 50 nm; lateral resolution < 150 nm), unrivaled geometric design freedom, and the rapidly expanding arsenal of reported functional materials make the technique highly attractive for microrobotics. To underline the versatility of TPL, it is worth mentioning that it can be employed as a 3D subtractive manufacturing technique provided the right photoresist formulation and degradation stimulus are used [40,41]. Similarly, it is possible to apply so-called positive photoresists, which function inversely by making the exposed material soluble, leaving the unexposed material behind after development [42].

In the early days of TPL, this fabrication technique found interesting applications where high resolution and accuracy were needed, such as in photonics [43]. Over the past decade, an impressive number of photoresist formulations comprising functional materials have become available [44]. This has opened up new possibilities for microrobotics, allowing the rapid prototyping of components such as sensors and MEMS [45]. The biggest drawbacks of TPL are the high equipment costs and the point-by-point method, similar to SLA, which renders it rather slow compared to projection techniques. To address the speed, Yi et al. recently showed a new method called optical force brush (OFB), which induces radicals to aggregate by optical forces. OFB further decreases the linewidth limit to 20 nm and provides substantial control over the linewidth, providing improved smoothness compared to the conventional layer-by-layer TPL [27].

### 2.5. Laser-Induced Forward Transfer (LIFT)

LIFT is a technique in which a pulsed laser forces a transfer from a donor ink layer onto an acceptor substrate. The donor layer can be either solid or liquid, and a wide range of materials, including metal nanoparticles and biomaterials, are suitable for this technique [22]. LIFT is mainly a tool for the rapid and accurate creation of patterns on the substrate, but it is also possible to use it to fabricate three-dimensional structures [21,34]. The patterning capabilities of this technique were first explored in OLED (organic light emitting diode) fabrication [46], but over the past two decades, it has been also applied to build sensors [47], electrochemical capacitors [48], and transistors [49]. While the wide range of materials makes LIFT potentially interesting for such a wide range of applications, there are some significant challenges ahead for use in microrobotics. Firstly, the requirement of a small distance from donor to acceptor is not always possible, or at least unpractical, in many applications, especially when the target has three-dimensional features [50]. Secondly, the poor adhesion strength of the transferred material is detrimental in situations where high conductivity is required. Solving these challenges and delivering on its promise to improve the already high resolution, LIFT has the potential to become a high-throughput fabrication technique for a great variety of microrobotic applications.

### 2.6. Selective Laser Sintering (SLS)

SLS is a printing technique that uses a laser to heat up a fine powder to the point where it usually combines/sinters below the melting point. By selectively repeating this process, a 3D structure is built layer by layer. SLS has several unique features compared to other fabrication methods such as the isotropic mechanical properties of printed parts, meaning that they have equal strength in all directions. Furthermore, SLS is suitable for the direct fabrication of both metals and polymeric materials. However, there are only a limited number of inks available because of the difficult challenge of producing powder of high quality (fine size, low size distribution, and consistent morphology). Mediocre power diminishes the resolution and leads to high porosity and thus low mechanical strength. Recent efforts to use plasma spheroidization of particles could reduce this drawback of SLS in the future by making high-quality powder production possible for a wider range of materials [51]. Even within the restrictions of available powders, no microrobotic applications have been reported with SLS fabrication, which can be explained by the poor resolution of most conventional systems [12]. These would show a resolution of 100–250 microns whereas the newest reported micro SLS systems reach a < 5 micron resolution, which is a great leap forward, although it will put even more stringent requirements on the powder quality [23].

### 2.7. Inkjet Printing

Inkjet printing uses diverse ink formulations that are deposited dropwise in a spatially selective manner followed by evaporation of the liquid. The small size of the droplets generated by piezoelectric pulses, usually between 3 and 100 pL, allows fast evaporation times that can effectively immobilize the target material in the desired position. Inkjet printing is an established technology to fabricate a variety of components in microrobotics, although the resolution is rather limited. Initially, it was used to build microlenses [52], but the scope has been expanded to sensors, actuators, and integrated MEMS systems [53]. This is a result of the simple use and the easy integration of multiple different inks including piezoelectric, conductive, and insulating materials [28]. Even though inkjet printing is a powerful tool in MEMS and microcircuitry fabrication, the inherent lack of control in depositing droplets is an almost insurmountable restriction to improve accuracy to a <10-micron resolution.

### 2.8. Microextrusion

Similar to inkjet printing, microextrusion is a bottom-up layer-by-layer printing technique in which material is pressed through a nozzle. However, in this case, there is no solvent involved and the pressure is constant, creating a continuous flow of deposited material instead of the inkjet droplets. The main advantages of microextrusion are its simple use, cost-effectiveness, mask-free execution, and wide range of suitable materials [30]. The initial applications involved microfluidics [54], but sensors and microelectronic devices [55] have been reported through the years. Although microextrusion is a very relevant technique for the larger elements or scaffolding for microrobotic fabrication, it is only applicable to low-resolution devices.

## 3. Emerging Microfabrication Technologies

The key parameters for the future of microfabrication are cost-effectiveness, scalability, resolution/accuracy, print speed, and material versatility (which we will discuss in the next chapter). Without exception, there is a trade-off between these hampering progress towards more complex devices and commercial applications. For instance, microextrusion is cost-effective and scalable but is lacking with regard to resolution, whereas 2PP has the exact opposite. We will discuss several recent developments, in contrast with the previous techniques not yet commercially available, that could change the field of microfabrication in the coming years.

### 3.1. Tomographic Volumetric Stereolithography (TVS)

The next generation of light-based 3D microfabrication aims to decrease printing time from hours to seconds by switching from 2D projection layer-by-layer printing to a rotating resist volume irradiated with a dynamically evolving light pattern [19,20]. The concept is based on the medical imaging technique called computed tomography (CT), which is used to construct a 3-dimensional image by combining images taken in a circle around the object. TVS inverts CT irradiating from multiple positions around the object and thereby constructing the structure. For the moment, improving the limited resolution is the most pertinent challenge in TVS, but it solves some of the issues of CLIP and DLP as it is specifically suitable for high-viscosity materials, is capable of combining multiple materials, and, above all, is much faster. A drawback of TVS could be the incompatibility with materials that have limited UV transparency or scatter the light.

### 3.2. Two-Step Absorption Polymerization

To make the high resolution achieved with two-photon lithography more widely available, it is essential to bring the equipment cost down. This is what Hahn et al. aimed to achieve with two-step absorption polymerization [31]. Similar to two-photon lithography, two-step absorption requires two photons to induce the cross-linking polymerization reaction. In TPP, a photoinitiator would absorb two NIR photons with a virtual intermediate state in between. In contrast, two-step absorption makes use of the resist formulation of a reluctant Norrish type I photoinitiator, e.g., benzyldimethylamine or dyacetyl, which, once excited, can undergo intersystem crossing to a triplet state that can absorb another photon to reach the triplet excited state and generate a radical. A quencher and scavenger molecule need to be present in the system as well to prevent inadvertent single-photon initiation. This system uses a 405 nm laser diode eliminating the need for the femtosecond NIR laser of TPP, one of the most expensive parts of the machine. The resolution of the two-step method is claimed to be equal if not superior to TPP.

### 3.3. Two-Color Two-Step

Another key parameter for future microrobotics is fabrication speed, which will always be challenging in point-by-point methods like SLA, TPP, and two-step polymerization. Alternatively, a light sheet covering an entire layer instead would be much faster. Unfortunately, methods like DLP are unsuitable to come even close to the fine resolution of TPP, as they require to replenish a new resist for each layer. The use of polymerizations initiated by dual-wavelength absorption phenomena (two-color two-step) could overcome this challenge by limiting the structure fabrication in a thin region of space defined by the intersection of a light sheet (of a specific wavelength) and a perpendicularly projected pattern (employing a different wavelength). For instance, volumetric polymerization has been reported by photoinitiation with blue light, while inhibiting it with a UV beam [56]. More recently, Hahn et al. took this sheet and projection setup while employing a similar resist system as the two-step absorption. The slight change compared to the two-step method was the use of a photoinitiator that absorbs light at different wavelengths in the triplet state [32]. A blue projection beam brought the molecules to the intermediate state, while a red sheet of light brought them to the triplet excited state initiating polymerization (Figure 3K). This approach can potentially achieve high resolution and high speed and could be a breakthrough for microfabrication. This is evidenced by the interest in bringing volumetric micro-3D printing from prototyping to serial production [57]. Furthermore, several new photoinitiators have been reported specifically for dual-wavelength applications [58,59].

### 3.4. Initiator Free Polymerization

All previously described lithography systems rely on photoinitiators to absorb laser light, induce polymerization, and thereby create a solid microstructure. Even though these chemicals are very effective, they have several drawbacks such as cost, discoloring of the system, and toxicity of the initiators themselves and their products after irradiation [33]. Especially in photonics and biomedical applications, a photoinitiator free-fabrication technique that does not suffer the same issues could be extremely beneficial. Several paths to overcome these challenges have been proposed, such as self-initiation polymerization by short-wavelength excitation of acrylates, which start absorbing efficiently between 220 and 260 nm [60,61]. A drawback of this method is the sensitivity to quenching by oxygen, which can be detrimental to the polymerization process. This is a well-known effect that also affects all of the previously described methods that utilize radical polymerization. However, the initiator-free system has a relatively low generation of radicals making the approach vulnerable to oxygen quenching. Currently, the PI-free microfabrication system has been reported, and a controlled oxygen environment might be required to build one. As an alternative to acrylates, it is possible to use a thiol-ene reaction without a photoinitiator, which has been reported for 266 nm SLA and 532 nm TPP systems [62]. A practical disadvantage of thiols is their pungent odor and toxicity.

## 4. Current Materials for Microactuation

To implement actuation in a microrobot, achieving control over the material shape change at the microscale is crucial. Several classes of materials suitable for actuation have been studied recently [14,44]. Here, we provide a fresh perspective by including the latest developments and comparing several virtues in Table 2.

### 4.1. Liquid Crystal Elastomers

Liquid Crystal Elastomers (LCEs) can provide high directional forces in response to light, heat, and electromagnetic fields, thus making them ideal candidates for directional actuation in untethered microdevices. In addition, their properties can be tailored easily by means of chemical synthesis and they are, in general, biocompatible [63,64,65]. Moreover, they are suitable for a wide variety of fabrication techniques including lithographic techniques, extrusion, and DLW, thus allowing the preparation of complex structures and functional architectures.

LCEs’ mechanical response relies on the ordering properties of liquid crystals. This class of materials usually comprises molecules (called mesogens) characterized by an oblong rigid part that tends to orient the molecular axis along a similar direction within a domain (called director). In this sense, these materials are able to show a high degree of ordering even when they are not in the solid state (i.e., mesophase). By cross-linking the aligned domains, the order is retained in the LCE. By heating above the phase transition temperature, this ordered phase is reversibly broken into a disordered random phase, causing a macroscopic deformation parallel to the director. Although there are several types of LC-ordered phases depending on higher-range ordering, for actuation applications, the nematic phase (which only has orientational and no positional order) is the most frequently employed.

The actuation of LCEs is strongly determined by the molecular ordering of the mesogens, which can be obtained by surface rubbing [66], mechanical stretching [67], electromagnetic fields [68,69], and polarized light [70]. The former is the most prominent ordering method in microrobotics and relies on the spontaneous alignment of the mesogens on a patterned surface before polymerization. This method limits the possible thickness (to approximately 50 microns), and thus, in planar alignment, the force output of an LCE structure, as the anchoring energies of surface interactions are finite. Guin et al. avoided this problem by laminating several layers of 50 μm thick LCEs and comparing their properties [66]. A four-layer laminate of these layers resulted in lifting 1100 times its own weight and a specific work capacity of 19 J/kg.

Molecular order can also be achieved by uniaxial stretching of a weakly cross-linked LC mixture, which is fixed by full cross-linking post-stretching. Kotikian et al. made 3D millimeter-sized structures by extruding LC mixtures orienting the mesogens in the printing direction and freezing structures by UV crosslinking [71]. On the microscale, Palagi et al. made rod-shaped microswimmers with a mechanical stretching method [72], but more complex microstructures would require incompatible technologies like DLW. Finally, electromagnetic fields are a suitable option for micromachines, which was shown by Yao et al. for magnetic fields [68]. Tabrizi et al. prepared millimeter-scale LCE structures using additive manufacturing where the orientation could be altered during the fabrication process using a magnetic field, showing the potential to spatially design the mesogen orientation using electromagnetic fields [69].

Additive manufacturing of LCEs usually relies on a photopolymerizable formulation comprising (meth)acrylates, which often require the design of special conditions or high-intensity light to avoid oxygen inhibition of the photo-induced radical polymerization. Different crosslinking chemistries can be employed to overcome this issue. For instance, McCracken et al. developed a liquid crystal with epoxy end groups enabling cationic photopolymerization (which is not sensitive to oxygen) and conveying the LCEs at a higher actuation temperature [73].

Photothermal stimulation is the current preferred method for LCE-based microrobot actuation as it allows for reversible and localized actuation with one single light source [30]. Indeed, at the microscale, a photothermal dopant allows for a rapid light-powered response and quick heat dissipation-driven return to the original shape [67,74]. Chen et al. used gold nanorods as photothermal fillers dispersed in LCE, allowing for NIR-stimulated actuation of 2PP written structures [75]. With a loading of 3 wt% of nanorods, the structure could contract approximately 20% in approximately 0.9 s when exposed to a 2 W laser. Light-absorbing small molecules can also perform as photothermal moieties. For example, a trans-isomer azodye can absorb light and store it as energy by changing to the cis-form; then, upon returning to trans, thermal energy is released in the surrounding matrix causing the LCE bulk to heat and deform. Martella et al. integrated within an LCE microgripper structure a 1 wt% azodye specifically designed to speed up the isomerization from cis to trans and red-shift the absorption of the dye to green light [10]. The design allowed for efficient actuation of the structures with a visible wavelength (with a response time of approximately 27 ms at a laser power of 51 mW). Recently, Montesino et al. reported an approach with two-wavelength stimulation allowing them to operate in the NIR [76]. They implemented a perylenebisimide dye, which absorbs green light, resulting not only in the conventional photothermal effect but also in the formation of new species with absorption bands in far-red and infrared. Irradiation of the latter leads to photothermal actuation, and, in addition, they regenerate the spectrum of the ground state dye. In this sense, one can employ green light to pattern the LCE reversibly and thereby obtain complex motion with NIR irradiation.

The same research group also prepared a microscopic artificial walker via TPP [77]. It consisted of four asymmetric IP-L legs connected to an LCE that could reversibly contract (at up to 2 kHz), which resulted in a directional force between the leg and the substrate and thereby movement. Similarly, Zuo et al. fabricated a walker, but in addition, they were able to control direction using different wavelengths [78]. The structure was prepared by laminated uniaxial stretched films comprising three different photothermal fillers, which absorbed different wavelengths, thus controlling the response in different areas and allowing forward and backward motion and steering. LCE-based microswimmers were developed in [72] (Figure 4), where a periodic light pattern induced peristaltic wave movement resulting in a net movement of the microstructure.

With swimmers, grippers, and crawlers being the current benchmark for LCE-powered microrobotics, the applications are limited to monodirectional photothermal actuation. This limitation is a result of surface rubbing being the common way to orient LCs, which has the limitation of having only one pattern, while spatially variable orientation could allow for much more complex actuation. To achieve this, other alignment methods have to be used on the microscale, for example, Yao et al. oriented mesogens using a magnetic field [68]. Tabrizi et al. used a bottom-up additive manufacturing technique in which a magnetic field (0.3 T) could be reoriented between every added layer [69]. Polarized light can be used to orient azodyes, which is already used for patterning surfaces [70], thereby indirectly orienting the LC. However, instead of this indirect approach, polarized light could be used in the future for direct spatial variation in the orientation of the LC in the 3D space, overcoming the thickness limitation of surface patterning. In a recent development, the group of Wegener and ourselves have approached this challenge with the use of electric fields [79,80]. Liquid crystal molecules with a large dipole moment orient themselves parallel to the direction of the field, enabling control of the direction during printing. The challenge to implement these orientation techniques in microrobotic manufacturing is essential to increase actuation complexity. LCE actuators are also limited by the actuation stimulus, which is usually (photo)induced heat. The previously mentioned multi-wavelength walker shows a development toward more sophisticated photoactuation [78]. Another example of additional complexity is a photothermal AND-gate LC network, reported by Lahikainen et al., which only actuates when illuminated by both UV and visible light [81]. As an alternative to light, there is the possibility of using electric field actuation or chemically induced actuation, which has not been shown for microrobots yet [82,83,84]. Another unexplored application of LCEs is its use as a passive material with variable Young’s moduli depending on temperature.

**Figure 4 micromachines-15-00275-f004:**
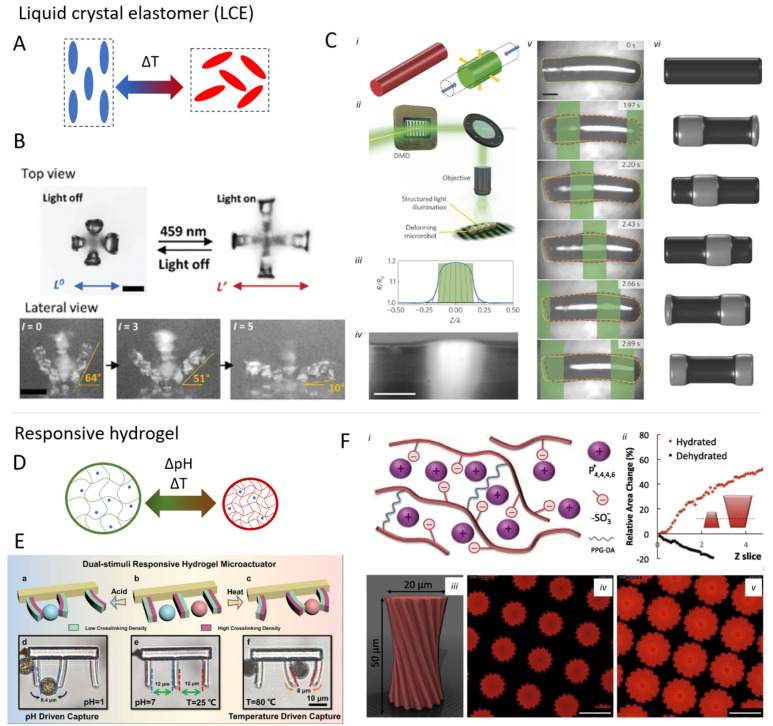
(**A**) Schematic actuation mechanism of liquid crystal elastomers (LCEs) that contract upon heating as the ordered molecular structure breaks. (**B**) Modification of a TPP-printed LCE actuator with various dyes allows for controlled actuation depending on the applied wavelength [85]. (**C**) Light-sensitive LCE micro-swimmers able to propel themselves via peristaltic movement triggered by green light [72]. (**D**) Schematic of hydrogel actuation, which depends on the absorption of water in the polymeric matrix. (**E**) Hydrogel microgripper sensitive to changes in both pH and temperature [86]. (**F**) Poly-ionic liquid hydrogel fabricated with TPP respond to temperature change based on the uptake of water [87]. (i) Structural model; (ii) shape variation upon swelling/deswelling; (iii) print model of the microstructures; (iv) microstructures as printed; (v) microstructures after immersion in water. Error bar in (iv) and (v) is 25 μm. Subfigure (**D**) is reproduced with permission from [86].

### 4.2. Stimuli-Responsive Hydrogels

An alternative material class to liquid crystals that does not require any orientation and can offer omnidirectional actuation is hydrogels. They consist of cross-linked hydrophilic polymeric materials that can form a gel upon absorption of water. They are compatible with additive manufacturing techniques and are particularly relevant for applications where biocompatibility is important, like tissue engineering. Hydrogel networks can swell and shrink based on their affinity with water or other polar solvents. By chemical design, one can prepare hydrogels that can respond to certain stimuli such as pH, ionic strength, light, temperature, and redox potential by changing their mechanical properties, viscosity, and shape [88,89,90]. The actuation is diffusion-controlled and therefore more rapid for small structures, making them an ideal material for nano- and micromachines. Furthermore, by dynamically changing the printing power in a fabrication technique like TPP, cross-link density gradients are formed, which can be exploited for the specific design of the actuation movement [91].

Hydrogel microstructures are prepared using UV curing followed by hydration. A structure can either be created by a mold [92], a mask [93], or other lithography methods including TPP [91] and stop-flow lithography [17] where only the latter two allow for 3D structures. Palleau et al. modified a macroscale hydrogel structure post-hydration by a technique called ionoprinting. This is a technique that can be potentially used in microrobotics and is based on the introduction of metal cations into a hydrogel by an electric field, which reduces the water content locally deforming the structure [94].

In this field of application, N-isopropyl-acrylamide (NIPAM)-based hydrogels are often encountered as they are characterized by a well-defined lower critical solution temperature (LCST) of approximately 32 °C, above which the poly-NIPAM becomes hydrophobic and dehydrates, allowing the hydrogel to reversibly swell/shrink upon heating. It is desirable that heat is wirelessly provided for microrobotics, which can be locally provided by a focused laser [91,95], but microactuators with light-absorbing dopants are also feasible, as Mourran et al. showed using PNIPAM doped with gold particles [92]. Other common triggers employed for the actuation of hydrogel microstructures include pH [96] solvent [97] and magnetic fields. For magnetic field actuation, the hydrogel is doped with magnetic particles, often iron (II,III) oxide. A more extensive discussion of magnetic actuation can be found in Section 4.3.

LCST-based actuation also offers an interesting platform to develop surface-bound microrobots. Hippler et al. used DLW to realize gray-tone printed poly-NIPAM cantilevers, which could actuate upon exposure to heat or laser in less than 100 ms [91]. The different degrees of cross-linking density allowed the authors fine control over the degrees of freedom during the actuation. Jin et al. further developed a photoresist to increase the deformation and built umbrella-like micromachines [98]. D’Eramo et al. used deep UV (250 nm) photopatterned poly-NIPAM for a microscopic cell-trapping system in which they could trap a single cell [99]. They also showed sealable microcages for isothermal RCA amplification. Rehor et al. prepared 100 µm-long poly-NIPAM crawlers using stop-flow lithography, which could move upon light stimulation thanks to friction hysteresis [95]. To improve the photothermal efficiency, they included gold nanoparticles in the blend.

As an alternative to poly-NIPAM, Wei et al. showed that is also possible to use pH-sensitive proteins such as bovine serum albumin (BSA) to prepare a responsive DLW-printed structure. They applied this in the preparation of a microsieve where the pore size could be directly controlled by changing the pH [100]. Similarly, Ma et al. employed a BSA-based photoresist to print via TPP pH-responsive microactuators on bendable microstructures [101]. Other biological materials can also be employed to prepare functional structures via TPP [102].

Polyionic polymers can actuate by a similar mechanism. The latter consists of polymeric chains bearing charged units balanced by counterions. They are generally softer than hydrogel, more flexible, and can incorporate more water resulting in a larger strain. Tudor et al. prepared via DLW microstructures of polyionic polymer, which could swell by 300% in water and showed LCST behavior, shrinking by approximately 30% upon heating. The LCST was found to be a wide range between 20 and 70 °C, giving precise control over the volume of the structure by temperature control [87].

On the macroscale, hydrogels have been designed to be sensitive to a variety of stimuli, but it is challenging to bring this down to the microscale. For instance, chemical stimuli are particularly suitable in the case of microdevices, as they permit orthogonal and controllable actuation, but in contrast to their macro counterparts, do not suffer the limitations in response time due to the diffusion mechanics. An example of this was reported by Ennis et al. employing sugar to rapidly actuate a boronic acid polymer [103].

In previous cases, swelling behavior is controlled by the diffusion of water in the hydrogel/PIL, while it is also possible to actuate using an electric field in an ionic polymer metal composite (IPMC). IPMC consists of a polyelectrolyte network or PIL with free-moving counter ions fixed between two electrodes. An applied electric field induces the migration of the counter ions, while the charges on the polymer are fixed, resulting in differential swelling of the IPMC gel and therefore bending. Reversibly, a mechanical input gives an electrical output that can be used in sensing applications [104]. For ion conduction, the water in the hydrogel should be saline, but alternatively, ionic liquids can be used, which have intrinsic ionic conduction. The main advantages of these so-called ionogels are their stability, low vapor pressure, low actuation voltage, and tunable ionic conductivity, but their response speed is lower. Research on IPMC actuation devices is not new, with an underwater millimeter-sized swimming robot already published in 2007 [105] and Hatipoglu et al. fabricating an ionic liquid IPMC micro cantilever using a focused ion beam in 2011 [106]. However, restrictions like low output force and non-standardized preparation steps have limited exploring its potential in applications on the micrometer scale, which is discussed in a compact review by Hao et al. [107]. On the macroscale, IPMC has already proven to be valuable in soft robotics, for instance, in a walking millimeter-sized robot [108]. An indication that IPMC actuators also have a future as additive-manufactured microactuators has been provided by Zhong et al. who developed thiol acrylate ionogel microstructures by masked photolithography but only showed the actuation on the macroscale [109].

Hydrogel microrobots have gained a lot of interest in the field of drug delivery, a field that might gain further traction in the future. The use of biochemical stimuli such as enzymes or sugars to build, for instance, a microrobot that releases insulin might be an attractive direction for the application of these types of materials. In general, the diversity in stimuli and dual material structures should be expanded in the future as also acknowledged by Rehor et al.; to direct motion in their crawler, they needed focused light, which required a complex set-up around the microrobot [95]. We can also expect new functionalities to the microrobotic hydrogels that are already used in macro-sized hydrogels.

### 4.3. Magnetic Polymers and Elastomers

Next to LCE and hydrogel-based actuators—which respond to light, heat, and chemical stimuli and find more applications comprising moving parts—magnetic systems cover a broad, somewhat complementary, spectrum of the research performed in microrobotics (Figure 5), focusing more on motion and spatial control. 

As organic materials do not show inherent magnetic properties, most magnetic microactuators consist of composites comprising magnetic particles (e.g., FePt, NdFeB, and Cr_2_O) or superparamagnetic iron oxide nanoparticles (SPIONs) dispersed in a soft matrix [112,113,114]. In this way, one can still take advantage of the common lithographic, molding, and printing techniques used in microfabrication, suffering only minor drawbacks connected to the different optical and rheological properties of the composites compared to the pure resins [100]. Alternatively, one may also prepare magnetic structures via the deposition of a thin layer of a magnetic metal like Ni or Co [115]. These limit the quantity of magnetic material employed in the microsystem, thus also affecting the force output, which is proportional to the total magnetization and the volume.

SPION-based composites are particularly interesting because the nano-size of the filler allows for a good and uniform distribution (even in microsystems) and because they are biocompatible. Our group highlighted these features in a series of publications several years ago [116,117,118]. More recently, the Sitti group employed SPIONs and natural polymers to prepare a series of magnetic actuators and helical swimmers via DLW to be employed in medical applications [111,119]. Sanchis-Gual et al. similarly fabricated helical swimmers but used DLW to build a micromold from PVA with iron oxide particles for magnetically actuating microrobots [120]. In general, we should mention that microswimmers and similar self-propelled devices for biomedical applications can often be prepared by competitive methods outside of additive manufacturing [121,122,123,124,125]. The future challenge for 3D fabrication lies in the integration of these techniques into complex tiny structures.

To supply microrobots with multiple functionalities, an interesting method is to combine responsive and functional materials with magnetic actuation. Wang et al., for instance, fabricated mm-sized bots employing a graphene-PDMS superhydrophobic composite comprising magnetic particles. Thanks to the hydrophobicity, the devices could move with less drag on water under the influence of a magnetic field or a NIR laser [126]. Magnetic field actuation is also particularly popular for drug-release microrobotics where hydrogel structures containing magnetic particles can be moved around using an electromagnetic actuation system. A second trigger causes drug release on a targeted spot. As a second trigger, Lee et al. used NIR light to shrink a poly-NIPAM spring-shaped hydrogel that releases a drug [127], while Li et al. used a poly(ethyleglycol-dicarylate) (PEGDA) gripper that opened upon pH change, releasing a drug [128]. Berger et al. also reported a hydrogel gripper with Fe_2_O_3_ particles, employing poly-NIPAM, which resulted in heat-responsive actuation [93]. Ceylan et al. prepared a magnetic microswimmer characterized by a double-helical architecture via two-photon lithography, which could move in response to a rotational magnetic field [111].

Recently, Pétrot et al. utilized magnetic actuation in a different but elemental approach [110]. Magnetic beads sized 30–50 µm were linked to a TPL-printed structure of only slightly larger size. The printed structure thus worked as an anchor for the magnetically sensitive bead. An applied magnetic field resulted in the reversible bending of this structure as the bead remained attached. In this way, the dynamic properties of tissues in vivo could be mimicked, making this a useful approach to studying cell behavior.

Besides actuation, the integration of magnetic particles in a photoresist can alternatively be employed as a sensor. Xu et al. constructed a microstructure capable of sensing forces of 0.5 piconewton with a printed spring system made out of PEGDA and 3 wt% iron-based superparamagnetic nanoparticles [129].

### 4.4. Electro-Active Polymers

Conductive polymers—such as poly(3,4-ethylenedioxythiophene) (PEDOT) blends, polypyrrole (PPy), and polyaniline—are also widely used as actuators, especially in soft robotics (Figure 6). The injection of charges in thin films of such materials or variation in their redox state can result in a volume change due to Coulombic repulsion between polymer chains or swelling by polar solvents and counterions, depending on the nature of the material and the environment. The stresses generated are in the order of tens of MPa [130]. In air, humidity desorption due to Joule heating can also lead to actuation [131]. Examples of micro-scale actuators are rather limited as they need to be supplied with electrical power from an external supply, which means they can be used in stationary microrobots but not in mobile, untethered ones [7,132]. However, miniaturization can improve the response time of such systems since they rely on diffusive events. Our group indeed showed that microfingers fabricated from ultrathin PEDOT:PSS films exhibit increased speed and output stress with respect to macro-sized structures [133].

Dielectric Elastomer Actuators (DEAs) are another common type of actuators in robotics, which requires electrical powering [134]. They make use of the attraction force that is generated between two charged capacitor plates to deform an elastomeric dielectric between them. As such, their actuation speed is extremely fast compared to the aforementioned systems (<1 ms), can deliver high forces (20 MPa), and their fabrication is rather straightforward using 2D techniques. Since the force output of a DEA is proportional to V2 and d-2 (where V is the applied voltage and d the dielectric thickness), such actuators can be reliable even at small scales. For example, Chen et al. employed them in a (tethered) cm-scale flying robot able to lift off and be controlled [135]. Yun et al. realized arrays of 200 µm thick PDMS-based DEAs, which they employed as a haptic element for skin sensation [136]. Remarkably, their fabrication method allowed the authors to orient the actuators perpendicular to the fabrication substrate. Recently, Kim et al. developed a methodology for the fabrication of DEAs using DLW and sputtered metallic layers, which they applied in the fabrication of actuating micromirrors and a crawling microrobot [137,138]. Such an approach is very promising for the realization of complex, miniaturized structures. However, DEAs need hundreds or thousands of volts to operate, which raises issues about their implementation in delicate environments (like the human body) and their motility in general (as they need wiring or batteries). Employing elastomers with a high dielectric constant, such as PVDF, can improve their force output and reduce the operating voltage by allowing more charge on the plates, but such materials are not compatible with many microfabrication techniques. Likewise, high-dielectric composites are not easily implemented because on the device scale, the dimensions of the filler are often not negligible.

## 5. Emerging Materials for Microactuation Technologies

It is no surprise that research in microrobotics and microactuation borrowed much from the biological system and from technologies used in soft robotics. However, there are many platforms from the latter whose full potential in microstructures has yet to be investigated. In the following paragraphs, we will explore those technologies that, despite being only marginally explored in microfabrication, still hold much potential for miniaturized devices and microrobots.

### 5.1. Shape Memory Polymers (SMPs)

An example of this can be found in shape memory materials, which are materials able to switch between two (or more) different preprogrammed shapes with temperature. Naturally, for actuation, one-way shape memory materials are not ideal as they cannot return to their initial form after the actuation event. Shape-change transitions in shape memory alloys have been reported to produce forces up to 200 MPa [63]. However, they usually require high temperatures (and high currents since they rely on Joule heating), which makes them impractical for most microrobotics applications. In comparison, Shape Memory Polymers (SMPs) have similar properties but show, for the applications discussed here, many advantages despite being capable of smaller forces (1 MPa): [139] (i) they can transition through several shapes (making them appealing for time-asymmetric motions); [63] (ii) the transition temperatures are smaller and tunable; and iii) their Young modulus is lower, and thus they comply better with other materials used in microfabrication. Moreover, many compositions could be suitable for 3D-printing technologies that can be used for both fabrication and imparting the shapes [140]. Jeske et al. described a photoresist formulation for the 3D microprinting of unidirectional SMP elements capable of storing up to 11 µJ of energy [141]. While programming the shapes at the microscale can be particularly difficult, one may expect that SMP composites able to respond to magnetic and electric fields could solve this issue, thus allowing fast parallel prototyping of functional, shape-memory micromachines. Zhao et al. provided a novel approach in this direction for shape-memory magnetic helical micromachines [142] (Figure 7). As in the case of the thermal actuators introduced so far (e.g., LCE, hydrogels), miniaturization can improve the response time of these systems since the heat capacity is proportional to the volume. Another method to program a shape or embed strain in a micro object is the use of holographic optical tweezers (HOT). Chizari et al. showed that it is possible to simultaneously TPP print and use tweezers by using a different wavelength laser (532 nm) for the HOT system. The authors could push polymerized material, which enabled the preparation of strain-embedded microsystems. This technology could give an extra degree of complexity to a microrobotic structure [143].

### 5.2. Self-Propelled Colloids

One thing that all the micro-systems introduced so far have in common is that to be used as propellers in Newtonian fluids such as water, they must produce a motion that is non-reciprocal, as described by the Scallops’ theorem for systems with low Reynolds numbers. This is possible, for example, by peristaltic or chiral movements [72,92], or by modifying the interactions between the microrobot and the substrate on which it is moving to obtain time-asymmetrical movements [77].

Another possibility is propulsion obtained by unidirectional production of gas or the generation of field gradients (phoretic self-propulsion) [144]. These kinds of propulsion usually rely on the catalytic decomposition of a high Gibbs free energy compound that acts as a fuel (most commonly catalytic decomposition H_2_O_2_) [145]. An asymmetric system where the resulting reaction products form predominantly on one side will experience a net push [146]. Electrophoresis, for instance, is achieved when the reaction happening on the particle surface creates a difference in charge distribution between the particle and the surrounding environment. These kinds of motors commonly exploit Janus architectures, where a nanoparticle is formed by two (or even more) materials segregated on the two different sides of the nanoparticle itself. Many different chemical reactions can be used, going from a rather simple dissolution reaction (where the fuel is on the motor itself) [147] to photo-activated ones (which allow control of the motor activity) [148,149,150]. It is worth mentioning that next to Janus particles, geometrical and fueling asymmetries can also result in directional motion [151,152].

While these latter systems are able to reach high speeds (30 µm s^−1^ in water), they require high fuel loadings not compatible with most environments and they lack maneuverability as they are not (yet) sensitive enough to small variations in fuel concentrations. This last issue can be addressed by the addition of another means of direction control, such as a magnetic core, thus obtaining devices able to efficiently propel and be easily controlled [153].

A rather new development for microswimmers is metal organic frameworks (MOFs) and covalent organic frameworks (COF). These highly porous materials provide a platform that can be exploited for drug delivery as well as propulsion [154]. For example, combining a photosensitizer with polypyrrole creates light-induced motion [155] (Figure 8B,C).

### 5.3. Molecular Motors

The 2016 Nobel Prize for nanomachines made Molecular Motors known to a broader scientific audience. Such compounds are molecules able to perform a unidirectional full rotation around a bond when exposed to light [158,159,160]. In this sense, by fixing them on a surface or on a particle, they can perform work by rotating in one fixed direction using light as a power source. One perk of these motors is that they can work both isolated and synergistically as assemblies, thus—in principle—providing a mean of actuation that can reliably power devices of several different dimensions, from the molecular to the millimeter scale [157,158] (Figure 8). Efficient nanomachines based on these principles are likely still 15–20 years in the future [161]; however, several examples showed how molecular motors can actively produce macroscopic actuation. The Feringa group demonstrated the possibilities offered by these systems in several examples, investigating, for instance, the ability of self-assembled, micron-thick strings of amphiphilic molecular motors to bend with light and lift a weight several times larger than their mass [162] or the properties of fiber aggregates of molecular motors in mimicking the extracellular matrix [156]. The Giuseppone group showed different examples of how, by covalently attaching molecular motors to flexible polymer chains, it is possible to fabricate materials that can shrink upon irradiation due to the entanglement of the chains prompted by the rotatory motion and that can reversibly return to the initial state upon application of an orthogonal stimulus or by employing the energy stored in the material itself [163,164,165]. In addition to Feringa’s motors, polymers comprising rotaxane-based motifs can also actuate under diverse stimuli [166].

### 5.4. Self-Assembled Materials and Microrobots

In addition to molecular machines, precise control over the interactions at the molecular level can also inspire a novel methodology for the design of highly specialized microrobots. Molecular self-assembly can be a powerful tool for bottom-up fabrication techniques, yielding regular domains capable of forming precise and responsive connections [167]. A widely studied approach in this regard consists of the use of DNA origami and other forms of DNA-based recognition and assembly [168] (Figure 9C). The intrinsic properties of the base pairs offer various advantages such as high selectivity to complimentary chains, remarkable strength of the supramolecular aggregates, and many functionalization possibilities to design particular patterns or tune specific interactions [169,170]. However, while this technology was very successful in the realization of functional nanometer-sized systems, it still suffers serious limitations when brought to the micron scale: as the system becomes bigger, there is a higher chance that defects due to the various local thermodynamic minima that may arise (i.e., local metastable structures) become incorporated in the self-assembled structures, thus affecting its whole stability and functionality; next to this, their response time is slow, usually higher than a second despite the small dimensions [170]. Nevertheless, there are examples of micron-sized devices based on DNA interactions [171]. For example, Maier et al. used DNA self-assembled microstructures to fabricate a series of biocompatible artificial flagella, approximately 10 µm long, which they could attach to magnetic microbeads, thus obtaining highly maneuverable, hybrid, magnetic microswimmers [172] (Figure 9B).

Besides molecular interactions, other forces can drive self-assembly and be effective even on larger scales, interfacing large particles as well as devices. Among such interactions, we find magnetic, dipolar, electrostatic, chemical, optical, and acoustic ones, thus displaying a plethora of opportunities to control the aggregation of particles and communication between them and allowing the simultaneous manipulation of large collections of microrobots [98,113,150,173,174,175,176]. Large swarms of microrobots can, in fact, be more suited to accomplish tasks that are too complicated for single microdevices, allowing for parallel processing and better task allocation. A recent review by Wang and Pumera investigated the means of interactions between micromachines in-depth and the arising behaviors [177]. Since it is still a challenge to integrate electronics into microdevices, physical and chemical interactions between microrobots (directly or indirectly through the environment) can generate novel approaches to achieve fine control and cooperation in an ensemble, also allowing us to rethink the design of task-oriented devices.

### 5.5. Multi-Responsive Materials

Although new technologies could undeniably bring novel approaches to the field—but might need time and resources to fulfill their potential—new and interesting solutions and functional micro-devices can arise by a combination of already known and established platforms as we already introduced previously in the discussion about SMP. This idea is not per se revolutionary, and one can find several recent examples in the literature. As examples, we already introduced the magnetic-electrophoretic micromotors prepared by Kline et al. [153]. Similarly, Chan et al. added a magnetic core to their Janus autophoretic motor, which moved upon dissolution of its Zn(OH) cover [147]. Fusco et al. and Ceylan et al. developed magnetic-driven carriers made of natural polymer composites loaded with anti-cancer drugs, which could be released only when decomposition occurred (triggered by pH or enzymatic action, respectively) [111,178]. Lee et al. even added a third trigger to their carrier, which could move magnetically, release drugs by pH, and change surface morphology upon altering the temperature [179]. This trend of multi-responsive actuation is also found in DLW fabricated actuators, as is evidenced by the recent work of Hsu et al. on LCE that deform differently upon irradiation with a different light wavelength [85] and by Wang et al. making hydrogel grippers responsive to both temperature and pH [86].

The incorporation of the same system with different actuation sources results in a synergetic effect that enhances functionality. In most of the examples we reported above, the magnetic response is widely used. This is because magnetic actuation is quite well understood, the fabrication methodologies are well established, and magnetic fields can reliably impart an orthogonal stimulus that does not interfere with any other actuation mechanism. Similar remarks can apply to acoustic stimulation, which, just like the magnetic one, can also reliably penetrate biological tissues in a non-invasive fashion, therefore making these technologies appealing for applications in the biomedical field. Indeed, orthogonality is a requirement that any actuator in a multi-responsive, multi-functional device must meet. In this regard, Aghakhani et al. combined acoustic propulsion and magnetic control in a DLW-printed microrobot that could move on flat surfaces with a speed in the order of millimeters per second, making them appealing for application in human vessels [180].

Many more combinations can be explored that can open up novel possibilities beyond the action-motion binomy. For instance, Kuksenok et al. demonstrated that a composite involving spatially defined photo- and thermo-responsive regions could generate motion with more degrees of freedom than the two alone [52]. Such composites, with different material phases on the microscale, can soon become a reality thanks to the development of multimaterial micro-3D printing techniques, therefore permitting the realization of functional architectures comprising several materials and micro composites commonly employed in soft robotics. Such systems could function as an on–off control of the activity, relying on the different mechanical properties of a thermo-responsive polymer. Besides combining different materials, the incorporation of various structures into an integrated platform could also generate a multi-responsive device. Our recent work on transfer printing of DLW structures should make integration, a previously daunting task, rather simple [181].

Among the various functional responses, the development of chemical control is still in its infancy. Hydrogels can respond to pH, ions, ionic strength, and redox potential, but typically in an on–off fashion. Instead, chemical phoretic propellers rely on high concentrations of fuel, which usually consists of strong oxidants or reductants. Synergetic interaction with other actuation mechanisms can help develop systems with a more gradual response, which can consume fuel more efficiently. Autophoretic microrobots comprising photocatalytic systems, for instance, can actuate only when irradiated by light [74] and could be good candidates as autonomous systems (if powered by solar light) for the photodegradation of pollutants in aqueous environments. In a recent study combining chemical and mechanical control, Huang et al. employed a magnetic hydrogel composite to prepare a series of compliant microswimmers, which could also change their shape in response to the osmolarity of the solution. In this way, they could finely adapt to the change in viscosity without suffering a drastic change in their maneuverability [182].

Another underdeveloped possibility for chemical actuation relies on the coupling of oscillating reactions with chemically responsive materials. Oscillating reactions are a class of reactions in which the concentration of the species involved does not change monotonically but reliably oscillates with time (with a period that depends on the temperature and the stirring) [183]. The variations in redox potential or pH that usually accompany the oscillation can therefore trigger a reversible response in a suitable material. The group of Yoshida reported a series of hydrogels comprising redox-active Ru(II) centers capable of undergoing rhythmic and peristaltic motions upon oscillatory redox potentials [184]. More recently, other groups reported conceptually similar systems based on different oscillatory reactions [183,185,186]. These systems offer good platforms to achieve rhythmic force outputs in the micrometer scale with one-time input (i.e., the initial loading of reactant), which could be used for autonomous actuation and propulsion mechanisms. However, for now, their working frequencies are rather small, and their applications are still limited by the diffusion of chemicals (reactants and products) in the environment surrounding the devices. After this, the biggest limitation of such systems might come from thermodynamics. As in any other chemical reaction, in a closed system, the free energy must decrease monotonically in time as the reaction proceeds, and oscillating systems are no exception. However, in the latter, the energy dissipation is very slow (despite being in far-from-equilibrium conditions), allowing for dynamic events to affect the relative concentrations of reactants and products [187]. As a result, while the total amount of work one can extract from a chemical oscillator is the same as that calculated by simple thermodynamic consideration of the system reaching equilibrium, the power associated with it would be drastically lower, thus raising several questions concerning the performances of such actuators in actual applications.

**Figure 9 micromachines-15-00275-f009:**
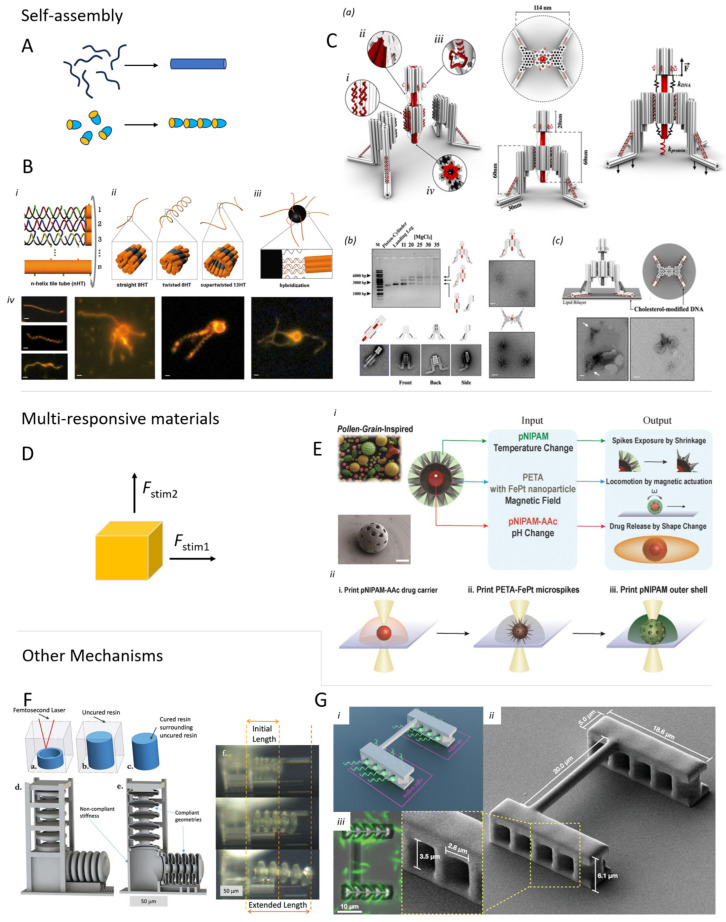
(**A**) Schematic representation of polymer or particle self-assembly to a structured conglomerate. (**B**) Self-assembled structure of DNA attached to magnetic microparticle, which can be propelled by means of a rotating magnetic field [172]. (i) DNA assembly schematic; (ii) design of different 8-helices; (iii) attachmento to magnetic bead (iv) Fluorescent microscopy pictures of different strucutres (scale bars are 1 μm. (**C**) Self-assembled DNA-based nano-winch in which the double helical DNA strings are schematically presented as cylinders (a). (b,c)TEM images show cholesterol-modified examples of the nano-winch legs [168]. (**D**) Schematic: multi-responsive materials respond depending on the type of trigger that is provided. (**E**) Schematic illustration of operation (i) and fabrication (ii) of a multi-functional microrobot. The structure comprises different materials providing magnetic steering, temperature-sensitive spikes, and pH-dependent drug release, respectively [179]. (**F**) Microrobot with uncured liquid in its core. Pumping additional liquid exerts pressure on the structure and induces hydraulic motion [188]. (a–c) Fabrication concept; (d–e) actuation mechanism; (f) Optical microscope pictures of the working device. (**G**) TPP-fabricated biohybrid robot that uses light-driven bacteria with a proton pump to propel and steer the 3D structure [11]. (i) model structure of the biohybrid device; (ii) SEM image of the vessel; (iii) interaction between the printed strucutre and the bacteria. Subfigure (**F**) is reproduced with permission from [188].

### 5.6. Pneumatics and Hydraulics

In macro-sized soft robotics, actuation is often achieved with chemically passive materials such as silicone. Motion in these cases is driven by either hydraulics or pneumatics, which provide fast and powerful actuation. Compared to material-driven actuation like hydrogels or LCEs, there would be no need for complex alignment set-up or cross-link density gradients, making the fabrication of pneumatics seem simple. In most microfabrication techniques, however, it is challenging to trap air inside the structure without destroying the product in a development step. The group of Reynaerts developed a molding technique in which a microrod was used to create a deep cavity in a cylindrical PDMS structure. They placed the structure on the pressure connector in a sequential step, which could induce bending of the structure as the hollowed-out part was not symmetrical. This resulting millimetric provided highly controllable and powerful actuation and was applied for a chip-on-tip endoscope, cilia biomimetic fluid propulsion [189,190,191]. Recently, our group evidenced the effectiveness of pneumatic actuation with a thermopneumatic electronic tattoo for tactile sensations [192]. Nevertheless, bringing pneumatics down to the scale of swimmers for vascular vessels might be unrealistic, but the potential of this approach could be very promising within the field of slightly larger MEMS and tethered microdevices. An alternative is hydraulics, which was applied by Smith et al. resulting in the effective bending of a DLW-printed structure, but the high viscosity of IP dip as the hydraulic fluid made motion rather slow with significant hysteresis [188] (Figure 9).

## 6. Perspective and Outlook

Ever since the first microrobots were developed, we have witnessed an ever-accelerating advancement of the technologies and materials involved in the field. In the first part of this review (and in Figure 1), we described how device design and fabrication methodologies had to constantly adapt to new disruptive technologies, materials, and insights. However, the emergence of rapid fabrication techniques with extremely high resolutions, mostly TPP, has been of singular importance in recent years. From this perspective, we have discussed which ideas have been key for progress and how the remaining challenges involving microrobotic functionalities (i.e., actuation, sensing, etc.) may be overcome in the future.

New technologies and materials will undoubtedly bring the field closer to widespread commercial applications and make it more accessible; however, the energy source will always remain a pivotal point for the design of a microrobotic device. Whether to use internal energy storage, energy leeching from the environment, or an external energy source (and what kind) is a decision that is made based on context, the available technology, and likely personal preference. Evidently, the choice of a certain solution has major consequences for the final functionality of a microrobotic device, which makes the debate both very interesting and important. Sitti and Wiersma have already provided an interesting comparison between magnetic and optical robots, pointing out several pros and cons. It seems inevitable that the field has to put effort and time into the discussion of the energy bottleneck [114].

### 6.1. Technology

Because fabrication technology is the basis for realizing devices, advancements in this area ripple through research fields to greatly influence material choices, device functionalities, and target applications. The emergence of rapid maskless prototyping techniques has led to the successful fabrication of increasingly complex microdevices. Compared to more traditional microfabrication methods, these novel techniques lend themselves better to rapid customization and prototyping. Furthermore, in the last decade, the use of soft materials has become mainstream as this has become more straightforward without the need for mask-based fabrication. In particular, two-photon lithography allowed researchers to fabricate complex microdevices, causing a significant leap forward in resolution and high flexibility. However, TPP-fabricated devices are not yet ready for commercialization. The clearest are the insufficient throughput and the high cost of the equipment. This could nevertheless change rapidly with new developments like two-step lithography and volumetric printing, which could drive down costs and increase speed dramatically. Commercial parties are already showing interest in the production of these new technologies. A further obstacle for these rapidly fabricated microdevices is their effective integration into commercial technological platforms such as MEMS and other microsystems, which is something that requires further studies as these technologies become more mature and appealing to the market.

The focus in this review has mainly been on the development of lithography-built soft devices, but advances in other technologies such as micro-SLS are not irrelevant as this allows for the use of other classes of materials. Combining different techniques and materials as well as working on various types of substrates is key to building integrated, complex microdevices.

### 6.2. Materials

Until recently, functional materials for high-resolution microfabrication techniques (i.e., TPP) were rare, but in the last decade, a wide range of actuating, sensing, and intelligent materials have been reported. Several classes of materials that were novel materials for microfabrication until recently have become entire subfields, and material development leads to increasingly larger control. For instance, liquid crystal elastomers as actuators and structural coloration elements have seen significant evolution in movement complexity, response time, and activation triggers thanks to developments in electric field orientation and its implementation in TPP. At the same time, an emerging trend is the aim of researchers to combine several functionalities that have different responses or integrate multiple materials in a single structure. The root of this trend lies in the desire to build microrobots with a high degree of control, for which multifunctionality in terms of propelling, visualization, sensing, and activity (e.g., drug release) is essential.

Among the more established materials like LCE, hydrogels, and magnetic composites, there are still new classes of materials introduced to the field of microactuation. Recently reported biohybrid, hydraulic, and improved high-dielectric composites are very relevant for the future of microfabrication. However, for commercial applications, there are other factors to be considered. For instance, standardization of printing conditions and resin formulation are essential to provide reliable and highly controllable device fabrication. It is therefore important for the field to show the value of each material development in actual microdevices. Generally, the use of soft materials poses challenges because of the low Young’s modulus (Table 2). This results in a lower quality factor (Q-factor) meaning that the devices have a relatively unstable output.

### 6.3. Emerging Methods

Some of the materials and technologies discussed in this review are still more of a scientific curiosity rather than a reliable platform for microrobotics. Self-assembly and molecular motors, despite their elegance from a chemical perspective and the analogies to the way biological organisms work, still lack the necessary power output to compete with current benchmarks. However, they have made immense progress in recent years and could have a huge impact on the field in the future, especially with the prospect of scaling down.

Another emerging method is the use of chemical control, which is applicable to many of the currently widely employed materials such as LCEs, hydrogels, and other responsive materials. The interesting perspective on using chemicals is the possibility of not just an untethered robot but rather an independent one that leeches off chemicals or solar power. The major challenge in this sense will be to find suitable systems that are able to work in ‘real’ environments, where the concentration of the fuel chemical could be small and the matrix effects could affect the efficacy.

In this review, we have largely ignored biohybrids because most of the technologies from Chapter 2 have only been applied to artificial systems. However, very recently, a fascinating microrobot was reported by Pelliciotta et al. They developed a method to exploit bacteria in a TPP-fabricated microstructure. The bacteria can absorb light to drive a proton pump with flagellar motors. This unique integration of biohybrids with the two-photon method resulted in light-induced self-propelled motion (Figure 9G) [11].

The future of rapid fabrication of microrobotics faces an interesting brink at the moment: for the first time, functional materials are reasonably available, complex design is possible, and new technologies seem to be able to improve throughput at lower costs. Real-life applications appear to be within touching distance albeit only for high-added-value devices for the near future. This means the most likely application target will be biomedical devices, which could involve robots within the human body that distribute medication, collect data, or obtain early detection of diseases like cancer. Outside our bodies, this target could involve skin devices, hearing aids, or smart lenses. All of these applications would require biocompatible, untethered, and complex microdevices.

## Figures and Tables

**Figure 2 micromachines-15-00275-f002:**
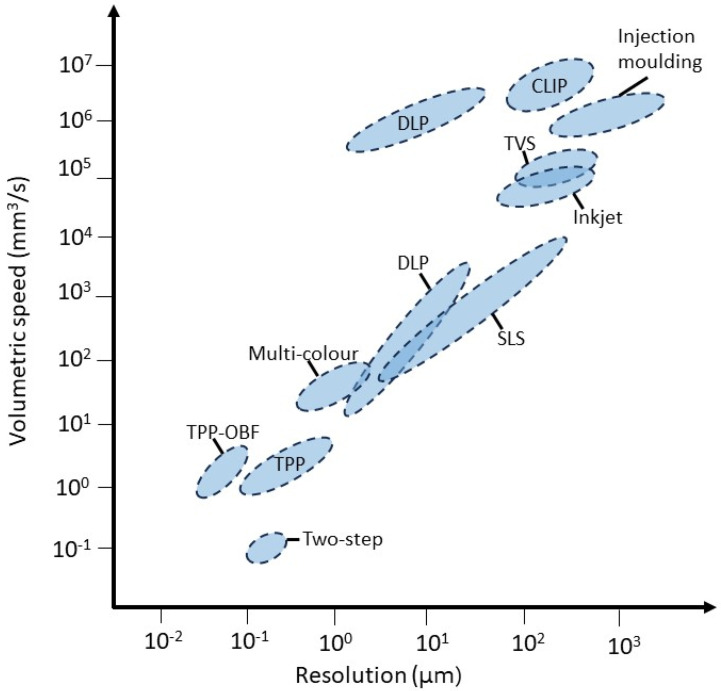
For each 3D rapid microfabrication technique, an approximate range of applicability in terms of resolution vs. printing speed is colored. The data in this figure are based on the values given in Table 1, which gives an approximation of the state-of-the-art resolution and speed for each technology. Obviously, other factors such as cost, practicality, and material suitability are just as important in choosing a certain technology for a particular application.

**Figure 3 micromachines-15-00275-f003:**
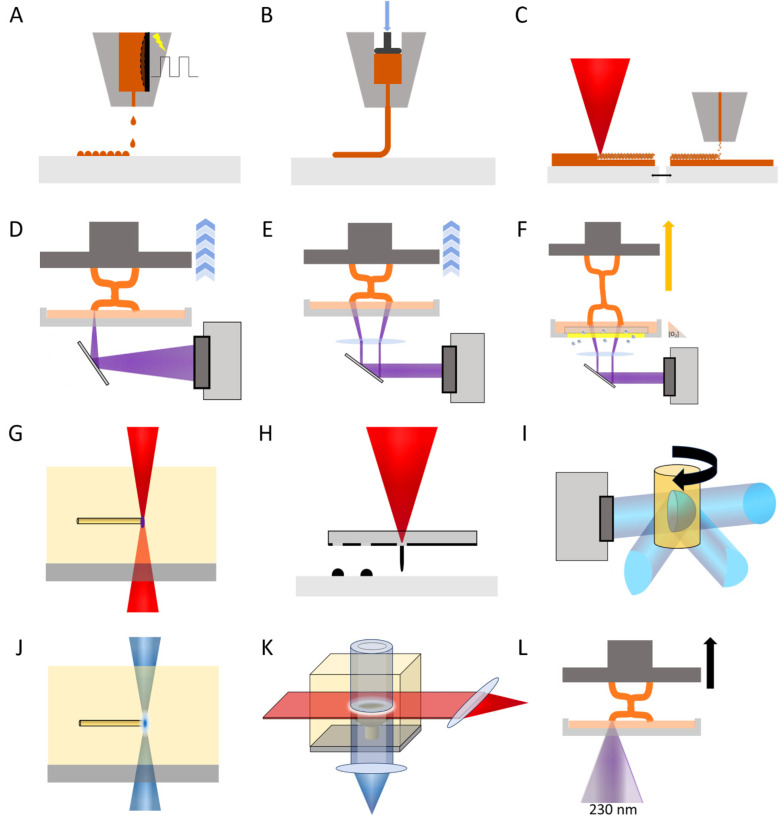
Schematic image of current and emerging technologies for 3D rapid microfabrication. If a laser is employed, a typical wavelength is given between brackets, bearing in mind that this is not absolute and can vary depending on the application. (**A**) Inkjet Printing; (**B**) Microextrusion; (**C**) Selective Laser Sintering (808 nm); (**D**) Stereolithography (365 nm); (**E**) Digital Light Processing (365 nm); (**F**) Continuous Liquid Interface Production (370 nm); (**G**) Two-Photon Lithography (780 nm femto-second laser); (**H**) Laser-Induced Forward Transfer (1064 nm); (**I**) Tomographic Volumetric Stereolithography (405 nm); (**J**) Two-Step Absorption Lithography (405 nm); (**K**) Multi-color Absorption Lithography (440 nm and 660 nm); (**L**) Photoinitiator-free Lithography (<250 nm).

**Figure 5 micromachines-15-00275-f005:**
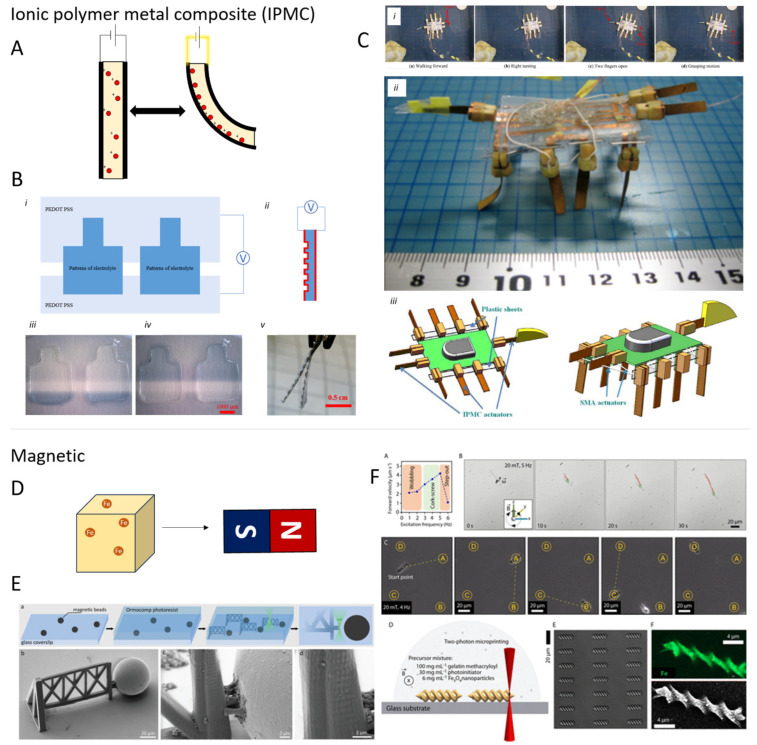
(**A**) Schematic representation of ionic polymer metal composite (IPMC) actuation upon electric stimulus. (**B**) Ionogel patterned with UV lithography for miniature IPMC actuator [109]. (i) Scheme of the actrive component; (ii) scheme of the device; (iii and iv) examples of color stimulation upon activation; (v) photograph that shows the actuation of the device. (**C**) Multi-leg millimeter-sized robot with IPMC beam cantilevers as legs that can walk controllably [108]. (i) Examples of motions; (ii) close-up photograph of the walker; (iii) schematics of the devices. (**D**) Schematic of magnetic sensitivity and actuation (**E**) A neodymium-iron-boron bead attached to a micrometer-sized printed structure, which can be actuated with a magnetic field [110]. (**F**) Helical microswimmer with superparamagnetic iron oxide particles functionalized in the hydrogel matrix. The microrobot can be actuated with a magnetic field (A, B, C), and drug release can be enzymatically triggered [111]. (**D**) Fabrication scheme; (**E**) printed microswimmers; (**F**) fluorescence pictures that show the ability of the hydrogel to absorb small molecule compounds.

**Figure 6 micromachines-15-00275-f006:**
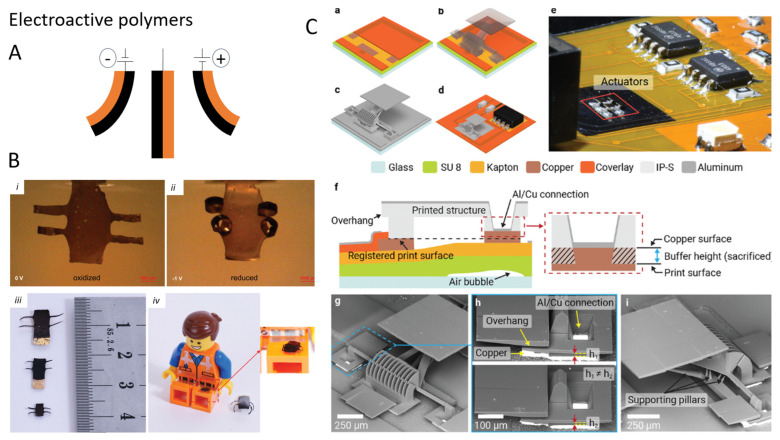
(**A**) Schematic representation of electroactive polymers. (**B**) 3D-printed microrobot with multiple arms that can be activated electrically [131]. (i) Image of the relaxed actuator; (ii) image of the contracted actuator; (iii and iv) images showing the possible different sizes of the microbot. (**C**) TPL-fabricated flexible microsystem that has an electroactively controlled array of micromirrors [130]. (a–d) Schematics of device fabrication; (e) (photograph of the actuators in an integrated circuit; (f) model section of the device; (g–i) SEM images of the actuators.

**Figure 7 micromachines-15-00275-f007:**
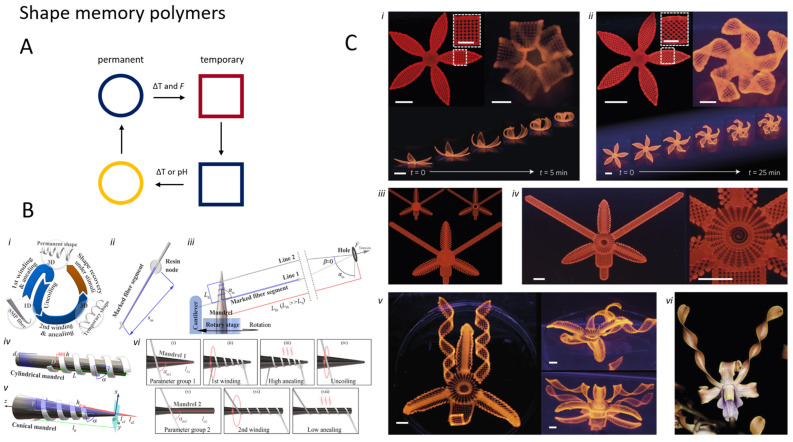
(**A**) Schematic representation of shape memory polymers (SMPs), which are mechanically programmed in a different shape at elevated temperatures. Upon a trigger like heat or pH, the shape returns to the original configuration. (**B**) Shape memory fibers are programmed and shown to undergo heat-induced shape recovery in helical microrobots [142]. (i) schematic of the working principle of the memry fibers; (ii and iii) fabrication procedure of the twiseted fiber; (iv and v) different mandrel shapes; (vi) fabrication examples. (**C**) Microsprings printed with TPP were compressed and cooled, and upon heating, they showed shape recovery releasing the stored elastic energy [141]. (i and ii) Examples of shape changes generated by the different printing patterns; (iii and iv) examples of shape recovery flower as prepared; (v) the recovery of the sahpe upon heating; (vi) comparison with flower. Error bars are 25 μm in the images and 10 μm in the insets. Subfigures (**B**,**C**) are reproduced with permission from [141,142].

**Figure 8 micromachines-15-00275-f008:**
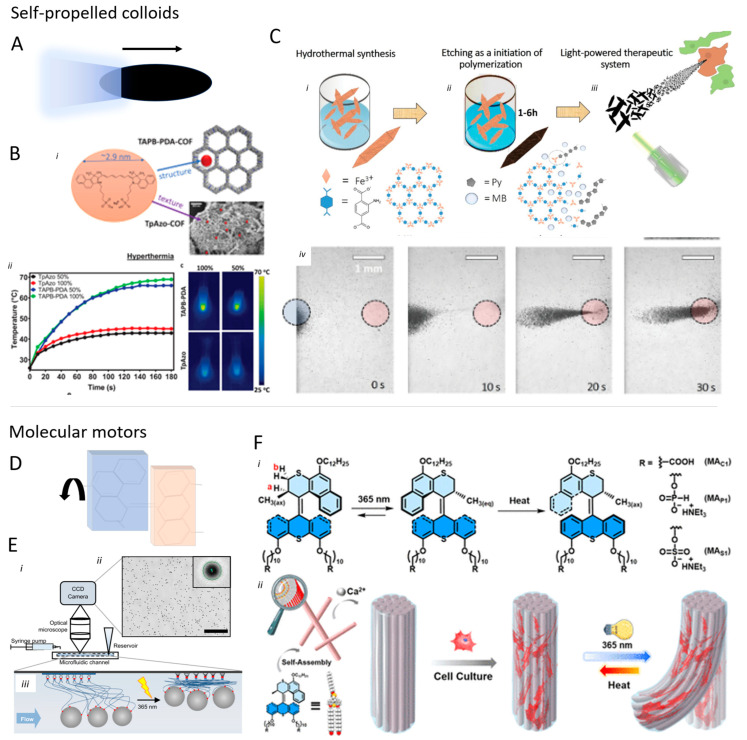
(**A**) Schematic representation of self-propelled colloids. (**B**) Covalent organic framework microswimmer propelled by light activation that is biocompatible. The pore size and large surface make it suitable for dyes for imaging or drugs for targeted release [154]. (i) MOF structural characteristics at different domains; (ii) temperature variation upon exposire time; (iii) visualization of the temperature variation on the device. (**C**) A polypyrole coating on a metal-organic framework with a methylene blue sensitizer enables light-propelled movement that is controllable with pH [155]. (i–iii) Fabrication of the microrobots (example of swarn movement. (**D**) Schematic representation of molecular motors. (**E**) Schematic illustration of supramolecular assembly that actuates in muscle-like manner on the macroscale, amplifying molecular motion [156]. (i) Scheme of the device; (ii) example of image; (iii) cartoon of the working principle. (**F**) Nanomotors/Molecular motors that can exert a pulling force on a cell membrane, which can potentially manipulate living cells [157]. (i) molecular motion in a molecular rotor upon light irradiation; (ii) example of assemblies that can act as a muscle to stimulate cells. SubFigure (**B**) is reproduced with permission from [125].

**Table 1 micromachines-15-00275-t001:** Overview of the current and emerging technologies for 3D rapid microfabrication with key characteristics, applicable materials, the lowest reported resolution, and highest reported printing speed.

Technique	Type	Suitable Materials	Resolution	Printing Speed	References
Stereolithography	UV-photopolymerization	Cross-linkable monomers, silicone, biodegradable materials	5–50 μm	Scanning: 100 mm/s	[15]
Digital light processing	Masked UV-polymerization	Cross-linkable monomers	1–50 μm	70–100 mm/h	[16,17]
Continuous liquid interface production	UV-polymerization with oxygen permeable window	Low viscosity photoresists	70	500 mm/h	[16,18]
Tomographic volumetric stereolithography	Dynamically evolving light beam around revolving resist	High viscosity, UV transparent photoresist	80–300	105 mm^3^/h	[19,20]
Laser-induced forward transfer	Pulsed laser forced transfer of ink	Metals, polymeric, biomaterial, hydrogels	10 μm	-	[21,22]
Selective laser sintering	Laser induced heating	Metal, ceramic, polynipam, polycaprolactone	5 μm	60 mm^3^/h	[23,24]
Two-photon lithography	Double NIR photon absorption	Photopolymerizable materials or positive resists	100 nm (20 nm with OFB)	Scanning: 100–625 mm/s	[25,26,27]
Inkjet printing	Droplets with dispersed nanoparticles deposition	Conductive, insulating piezoelectric, polymeric	50 μm	Scanning: 8 m/s	[28,29]
Micro-extrusion	Continuous material deposition	Hydrogels, polymeric materials, PDMS	150 μm	5 mm/s	[30]
Two-step absorption	UV-polymerization	Photopolymerizable materials	100 nm	Scanning: 4 mm/s ^1^	[31]
Multi-color two step	Volumetric printing by double beam activation	Photopolymerizable materials	0.5–1 μm	7 × 10^6^ μm^3^/s	[32]
PI free polymerization	Deep-UV absorption by acrylates	Acrylates	-	-	[33]

^1^ As an emerging technology, the current state-of-the-art maximum speed is rather low and not representative of its potential.

**Table 2 micromachines-15-00275-t002:** Overview of material classes for microactuation and their actuation characteristics including the stress generated, Young’s Modulus, response time, potential actuation stimuli, and the type of actuation. The bottom part of the table describes materials that have not yet been reported in microrobotics but should be suitable for future use.

Material	Stress Generated	Young’s Modulus	Response Frequency	Activation Stimuli	Actuation Type
LCE	0.01–0.1 MPa	1–80 MPa	1–100 Hz	Heat Light Swell Chemical	Anisotropic shape change
(ionic & non ionic) Hydrogel	0.1–10 MPa	0.01–5 MPa	1–10 Hz	Heat Light Swell Redox pH chemical Electric field	Isotropic shape change
Magnetic composite	Dependent on the filler	Dependent on the matrix	(20 µm s^−1^) (1–10 Hz)	Magnetic field	Independent motion, (multi-)directional shape change, rotatory motion
Auto-phoretic micro motors	---	200–300 GPa	(20 µm s^−1^) (0.1–0.5 Hz)	Chemical Light Redox	Unidirectional propulsion, rotatory motion
**Not yet established in microrobotics**					
(two-way) SMA	200 MPa	200 GPa	Up to 100 Hz	Heat (Joule heating)	Directional shape change
(two-way) SMP	0.1–1 MPa	20–50 MPa	<1 Hz	Heat	(multi-) directional shape change
Molecular motors	(3 kW mol^−1^)	---	<10 Hz	Light	Rotatory motion
DNA self assemblies	---	---	<1 Hz	Light chemical	Bending motion Anistropic shape change
Electrode + Material systems					
Conductive polymers (component)	10–50 MPa	---	Up to 1 kHz	Charge Current Redox	Bending motion
Dielectric Elastomer Actuators (component)	20 MPa	---	Up to 10 kHz	Electric field	Anisotropic shape change
Ionic polymer actuators (component)	---	---	<10 Hz	Electric field	Bending motion
Hydraulic	---	---	----	Pressure	Shape change (extension)

## Data Availability

No new data were created or analyzed in this study. Data sharing is not applicable to this article.

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
