# Peer review of "Evolution of the Microrobots: Stimuli-Responsive Materials and Additive Manufacturing Technologies Turn Small Structures into Microscale Robots"

_micromachines, 2024, doi:10.3390/mi15020275_

Round 1
Reviewer 1 Report
Comments and Suggestions for Authors
This review paper systematically summarizes recent advancements in the field of microrobots, with a focus on smart materials and fabrication methods. Although there are already some published papers on a similar topic (e.g., "Fabrication and optical manipulation of micro-robots for biomedical applications, Matter 5, 3135–3160"), the current review maintains a forward-looking perspective, particularly in the discussion of photopolymerization technologies. Overall, the manuscript is publishable after major revisions. The following issues should be considered to further enhance its quality:
1. On Line 130, Page 5, the author demonstrated that stereolithography(SLA) could be used for exposing positive photoresists. While, typical SLA is based on liquid-state negative resins, (e.j. A review on stereolithography and its applications in biomedical engineering, Biomaterials, 31 (2010) 6121e6130.) The exposure principles of solid-state positive photoresists may differ from traditional SLA.
2. On Line 170, Page 6, the term "Two-photon lithography (TPP)" is incorrect; it should be expressed as "two-photon polymerization (TPP)" or "two-photon lithography (TPL)."
3. In the section titled “Current Fabrication Technologies for microrobotics”, the author seems to primarily discuss additive manufacturing methods more precisely printing methods. Nevertheless, varied advanced methods such as subtractive manufacturing and self-assembly processing methods in microrobot fabrication have been overlooked. To match the scope of the article, the author should supplement more processing techniques or use a more appropriate title like "Current Printing Technologies for Microrobotics" instead of "Current Fabrication Technologies for microrobotics".
4. In Figure 3, the methods involving laser processing should explicitly indicate typical laser wavelengths more than color difference.
5. In Table 2, if the author defines "Response time", it should be presented in time units (e.g. s/min/h) rather than frequency.
6. In the section titled “Emerging microactuation material technologies”, the author only provides polymer-based materials for microrobots. However, to my best knowledge, various inorganic materials such as silicon, nickel, and graphene could also be utilized for fabricating microrobots. In this scenario, some biocompatible materials such as protein-based robots should also be discussed as they are highly programmable and proteins/ hydrogels are also biocompatible for the construction of biohybrid mocrorobots. (e.g., Femtosecond laser programmed artificial musculoskeletal systems, Nature Communications, (2020) 11:4536; X. Wang, D. Lin, Y. Zhou, N. Jiao, S. Tung, L. Liu, ACS Nano 2022, 16, 14895. ) As this article focuses on advanced materials, the author needs to enrich the corresponding content in this section.
7. The section titled "Emerging microactuation material technologies" is not appropriate for the subsequent discussion of SMPs and multi-responsive materials. In my opinion, materials are not a type of technology; materials refer to substances, while technologies are the strategies of processing them.
8. In the "Perspective and Outlook" section, the author briefly discussed self-assembly as an emerging method. This representative technology is suggested to be discussed more profoundly within the section on "Current Fabrication Technologies for microrobotics."
Comments on the Quality of English Language
Engligh needs minor improvement.
Author Response
Reviewer #1: This review paper systematically summarizes recent advancements in the field of microrobots, with a focus on smart materials and fabrication methods. Although there are already some published papers on a similar topic (e.g., "Fabrication and optical manipulation of micro-robots for biomedical applications, Matter 5, 3135–3160"), the current review maintains a forward-looking perspective, particularly in the discussion of photopolymerization technologies. Overall, the manuscript is publishable after major revisions. The following issues should be considered to further enhance its quality:
We thank the reviewer for the kind and encouraging feedback.
- On Line 130, Page 5, the author demonstrated that stereolithography(SLA) could be used for exposing positive photoresists. While, typical SLA is based on liquid-state negative resins, (e.j. A review on stereolithography and its applications in biomedical engineering, Biomaterials, 31 (2010) 6121e6130.) The exposure principles of solid-state positive photoresists may differ from traditional SLA.
R: We understand the confusion this sentence might cause to the readers and thus, such methodologies being not common in SLA, we decided to remove it. Instead, we have added an example of the use of a positive photoresist to the TPL section where it is a useful methodology.
- On Line 170, Page 6, the term "Two-photon lithography (TPP)" is incorrect; it should be expressed as "two-photon polymerization (TPP)" or "two-photon lithography (TPL)."
R: We thank the reviewer for pointing out this irregularity. We have chosen to consistently use the term “two-photon lithography (TPL)”.
- In the section titled “Current Fabrication Technologies for microrobotics”, the author seems to primarily discuss additive manufacturing methods more precisely printing methods. Nevertheless, varied advanced methods such as subtractive manufacturing and self-assembly processing methods in microrobot fabrication have been overlooked. To match the scope of the article, the author should supplement more processing techniques or use a more appropriate title like "Current Printing Technologies for Microrobotics" instead of "Current Fabrication Technologies for microrobotics".
R: We thank the reviewer for this helpful suggestion. We agree that a more specific title will clarify the scope of the section and therefore we changed the title to "Current Additive Manufacturing Fabrication Technologies for microrobotics”.
We agree with the reviewer that there is a wide variety of manufacturing methods for microrobotics beyond the selection that we discuss in the review. Self-assembly has shown interesting potential as a segment in a multi-step fabrication top-down process, but we do not believe that an approach as such is mature enough to produce microsystems with functionalities and design freedoms comparable to those discussed in other parts of this review. However, we do have a similarly positive outlook with the reviewer on the prospect of self-assembly, which is why we discussed it in the emerging section of the review (previously section 4.4, but after revision moved to section 5.2).
We agree with the reviewer that also subtractive manufacturing is a valuable method that can be applied with various lithographic techniques. Therefore, we have addressed subtractive manufacturing the TPL section and added relevant reports with fabrication principles that can be employed within other technologies.
- In Figure 3, the methods involving laser processing should explicitly indicate typical laser wavelengths more than colour difference.
R: We have specified the wavelengths in Figure 3. However, it is worth mentioning that, from a practical perspective, several different lasers (characterized by similar wavelengths) can be used to obtain similar results.
- In Table 2, if the author defines "Response time", it should be presented in time units (e.g. s/min/h) rather than frequency.
R: We thank the reviewer for pointing out this discrepancy. We found comparison of reported values to be easier in frequency units and thus changed the title to “response frequency”.
- In the section titled “Emerging microactuation material technologies”, the author only provides polymer-based materials for microrobots. However, to my best knowledge, various inorganic materials such as silicon, nickel, and graphene could also be utilized for fabricating microrobots. In this scenario, some biocompatible materials such as protein-based robots should also be discussed as they are highly programmable and proteins/ hydrogels are also biocompatible for the construction of biohybrid mocrorobots. (e.g., Femtosecond laser programmed artificial musculoskeletal systems, Nature Communications, (2020) 11:4536; X. Wang, D. Lin, Y. Zhou, N. Jiao, S. Tung, L. Liu, ACS Nano 2022, 16, 14895. ) As this article focuses on advanced materials, the author needs to enrich the corresponding content in this section.
R: We thank the Reviewer for their comments. The papers they suggested and other have been added to the relevant sections and discussed (in particular in section 4.2 ‘Stimuli-responsive hydrogels’ and 4.3 ‘Magnetic polymers and elastomer’. We did not want to dwell much on these topics as we wanted to place our focus more on stimuli-responsive materials that can act as controllable actuators (indeed, in the title we refer to microactuation specifically, not to microrobotics in general). Some examples hydrogel materials that fit this purpose were proposed in the appropriate section and we stated in the introduction that biohybrids are not in the scope of this review. Concerning silicon, we decided not to include it as we believe that it is an already established material and overlapping much with the fields of MEMS. In this sense, one can find many reviews and examples that address its use and implementation in the fabrication of microdevices. We made these ideas clearer in the Introduction.
- The section titled "Emerging microactuation material technologies" is not appropriate for the subsequent discussion of SMPs and multi-responsive materials. In my opinion, materials are not a type of technology; materials refer to substances, while technologies are the strategies of processing them.
R: We thank the reviewer for this relevant distinction. We understand the ambiguity of the terms employed and therefore we have changed the title to "Emerging materials for microactuation technologies".
- In the "Perspective and Outlook" section, the author briefly discussed self-assembly as an emerging method. This representative technology is suggested to be discussed more profoundly within the section on "Current Fabrication Technologies for microrobotics."
R: We thank the reviewer for this suggestion. The main focus of microrobotic manufacturing technology in the manuscript was additive manufacturing, which the method to build microrobotics with the most extensive design freedom. As mentioned earlier, we have dedicated subsection 5.3 to self-assembly as we agree that it is a powerful in microfabrication, however, we do not believe that the method has matured sufficiently to be compared to other fabrication technologies that are more ubiquitous and therefore available to a broader audience. In this regard, additive manufacturing is in a much further stage of development making any comparison difficult and unfair.
Reviewer 2 Report
Comments and Suggestions for Authors
This review intends to provide an overview of the major fabrication technologies and advanced materials used to fabricate microrobots in the past 1 or 2 decades. The manuscript also provides an interesting outlook on the prospects of emerging fabrication technologies and materials. The manuscript is well-written. The sections are well-organized. However, the specific works being reviewed seem to contradict the scope of the review. Overall, the content of the review does not fully align with the scope that the authors claim to cover. Please see the comments below for details:
1. In section 2 “Current Fabrication Technologies for microrobotics”, is the intent to focus only on 3D rapid prototyping techniques? Off the top of my mind, I can think of a few widely used microfabrication techniques for mobile microrobots not included in your list, such as GLAD, photolithography, electrodeposition, self-scrolling, etc. It is not so clear because Table 1 and Figure 1 are limited to 3D prototyping, but the main text here (line 97-110) seems to include the “realm of microfabrication”, which should include the techniques mentioned. Please clarify in your manuscript regarding the scope of the fabrication technique being reviewed.
2. If the scope is limited to 3D rapid prototyping, please have a good justification, while the microfabrication techniques I mentioned are not classified as 3D rapid prototyping had been used quite successfully to fabricate microrobots with a variety of materials, which seem to fit within the scope of this review.
3. As the authors mentioned, the key parameters are “cost-effectiveness, scalability, resolution/accuracy, print speed and material versatility”; if this review is not restricted to 3D rapid prototyping, the techniques I mentioned in comment 1 are quite competitive in terms of these 4 parameters. For example, GLAD can mass-produce helical propellers from micro to nanoscale, is relatively cost-effective, and can incorporate different types of material (10.1002/adma.202001114). Photolithography can be used to make planar achiral microswimmers using photoresist (10.1021/acsami.2c18955), MOF (10.1021/acs.iecr.1c01409), and hydrogel (10.3389/fbioe.2023.1086106). Photolithography can also be used to mass-manufacture electronically integrated microscopic robots with voltage-controlled actuators called SEAs (10.1038/s41586-020-2626-9). Lithographic techniques combined with self-scrolling were able to create helical microswimmers (10.1063/1.3079655) and soft deformable micromachines (10.1038/ncomms12263). Template-assisted electrodeposition can make helical microrobots (10.1002/smll.201302856). These are just some of the work that comes to my mind (there are more).
4. I think the soft microrobot with super-compliant picoforce springs is worth mentioning (10.1038/s41565-023-01567-0). It uses TTP and acrylic elastomer photoresist to crease the picoforce springs on mobile microrobots, and then the springs actuate the microrobots to move forward. It seems quite fitting for this review.
5. Figure 4 caption, (b) and (c) should be switched.
6. Some of the materials reviewed here do not fit the scope (as I understand it) of this paper. The fabrication section of the manuscript reviewed 3D rapid prototyping technologies. However, some of the materials mentioned in sections 4 and 5 were not used with 3D rapid prototyping. For example, the microswimmers in ref 124 ( line 619 and Figure 6(b)) were fabricated using chemical synthesis, which was not discussed in sections 2 and 3. Another example is the microswimmers created using the self-assembly of DNA tiles. There are a lot of examples like these in the manuscript, like many of the references from sections 4.4 and 5.3. Thus, there seems to be a major disconnect between the fabrication technique and some of the material being reviewed here. The authors even stated that “we intend here to provide an updated overview of the main materials and applications connected to the techniques”. I suggest expanding the scope of the fabrication techniques, maybe adding another section to review the fabrication techniques used with the materials mentioned in sections 4 and 5.
7. Table 2, “DNA self assemblies” was placed in the category of “Not yet used in microrobotics”. But ref 156 shows that DNA was self-assembled into microswimmers.
Author Response
Reviewer #2: This review intends to provide an overview of the major fabrication technologies and advanced materials used to fabricate microrobots in the past 1 or 2 decades. The manuscript also provides an interesting outlook on the prospects of emerging fabrication technologies and materials. The manuscript is well-written. The sections are well-organized. However, the specific works being reviewed seem to contradict the scope of the review. Overall, the content of the review does not fully align with the scope that the authors claim to cover. Please see the comments below for details:
R: We thank the reviewer for the kind compliments and constructive feedback.
- In section 2 “Current Fabrication Technologies for microrobotics”, is the intent to focus only on 3D rapid prototyping techniques? Off the top of my mind, I can think of a few widely used microfabrication techniques for mobile microrobots not included in your list, such as GLAD, photolithography, electrodeposition, self-scrolling, etc. It is not so clear because Table 1 and Figure 1 are limited to 3D prototyping, but the main text here (line 97-110) seems to include the “realm of microfabrication”, which should include the techniques mentioned. Please clarify in your manuscript regarding the scope of the fabrication technique being reviewed.
R: We thank the reviewer for bringing up the ambiguity of the scope. Our intention was to focus on 3D structures but we agree that the message could not have come across. To address this issue – also raised by Reviewer 1 –, we have made several changes in the manuscript. More relevantly, we changed the title to mention explicitly additive manufacturing and stimuli responsive materials, and also parts of the abstract, introduction and section 2 to be more explicit about the scope.
In this sense – and for the sake of the length of the manuscript –, as we recognize the importance of the mentioned fabrication methods, the technologies suggested by the reviewer were only briefly mentioned as alternatives for AM in section 2.
- If the scope is limited to 3D rapid prototyping, please have a good justification, while the microfabrication techniques I mentioned are not classified as 3D rapid prototyping had been used quite successfully to fabricate microrobots with a variety of materials, which seem to fit within the scope of this review.
R: We thank the Reviewer for pointing out this crucial point. We believe that the focus on 3D additive manufacturing is warranted by its unmatched design freedom in shape and stimuli-responsive polymeric materials. To achieve locomotion or other complex microactuation tasks, 3-dimensional freedom offers superior possibilities and design flexibility. Furthermore, the relative novelty of the 3D methods means that they have opened up many new and exciting possibilities that were until recently unfathomable.
We tried to clarify this view through several modifications in the introduction and the rest of the manuscript to better justify the scope.
- As the authors mentioned, the key parameters are “cost-effectiveness, scalability, resolution/accuracy, print speed and material versatility”; if this review is not restricted to 3D rapid prototyping, the techniques I mentioned in comment 1 are quite competitive in terms of these 4 parameters. For example, GLAD can mass-produce helical propellers from micro to nanoscale, is relatively cost-effective, and can incorporate different types of material (10.1002/adma.202001114). Photolithography can be used to make planar achiral microswimmers using photoresist (10.1021/acsami.2c18955), MOF (10.1021/acs.iecr.1c01409), and hydrogel (10.3389/fbioe.2023.1086106). Photolithography can also be used to mass-manufacture electronically integrated microscopic robots with voltage-controlled actuators called SEAs (10.1038/s41586-020-2626-9). Lithographic techniques combined with self-scrolling were able to create helical microswimmers (10.1063/1.3079655) and soft deformable micromachines (10.1038/ncomms12263). Template-assisted electrodeposition can make helical microrobots (10.1002/smll.201302856). These are just some of the work that comes to my mind (there are more).
R: We thank reviewer for these points and we agree that the mentioned techniques are quite relevant. As the comments we received pointed out, the realm of microfabrication is immense and for that reason we decided to focus on AM technologies usually associated with fast prototyping. We believe that 3D AM allows for unmatched versatility in terms of 3D shapes and use of advanced functional materials. We apologize if such scope was not clear in the first version of the manuscript and we hope that the changes we made to address the other points of this report are sufficient in doing so. In any case, because of their relevance, some of the references suggested by the Reviewer were added to the manuscript.
- I think the soft microrobot with super-compliant picoforce springs is worth mentioning (10.1038/s41565-023-01567-0). It uses TTP and acrylic elastomer photoresist to crease the picoforce springs on mobile microrobots, and then the springs actuate the microrobots to move forward. It seems quite fitting for this review.
R: We thank the review for this interesting addition. The suggested paper was not yet published at the moment of first submission. We have included it in the revised version.
- Figure 4 caption, (b) and (c) should be switched.
R: Thank you, we have corrected this mistake.
- Some of the materials reviewed here do not fit the scope (as I understand it) of this paper. The fabrication section of the manuscript reviewed 3D rapid prototyping technologies. However, some of the materials mentioned in sections 4 and 5 were not used with 3D rapid prototyping. For example, the microswimmers in ref 124 ( line 619 and Figure 6(b)) were fabricated using chemical synthesis, which was not discussed in sections 2 and 3. Another example is the microswimmers created using the self-assembly of DNA tiles. There are a lot of examples like these in the manuscript, like many of the references from sections 4.4 and 5.3. Thus, there seems to be a major disconnect between the fabrication technique and some of the material being reviewed here. The authors even stated that “we intend here to provide an updated overview of the main materials and applications connected to the techniques”. I suggest expanding the scope of the fabrication techniques, maybe adding another section to review the fabrication techniques used with the materials mentioned in sections 4 and 5.
R: We thank the reviewer for identifying this issue between the scope of fabrication technologies and the described materials. We have made several changes to the manuscript in hope of staying within the boundaries of the set scope.
As mentioned earlier, the intended scope of the review is to provide a brief overview of 3D additive manufacturing technologies and employed responsive materials of interest of microrobotics. Compared to other reviews in the field, our manuscript wants to extend toward the future of both technology and materials, critically assessing the fabrication and chemical technologies that could change the paradigm of the future in the next years. We believe that these “emerging” parts (section 3 and 5) allow the reader to think critically about the direction of the field. Part of this scope is to discuss materials that are particularly promising in impacting in the future and change paradigms. For instance, we are not aware of self-assembly as a bottom-up process to make a fully functional microrobot at the present time, but self-assembled parts might be successfully implemented in 3D architectures in the future.
We acknowledge that we should have placed section 4.4 in the “emerging” section (5). We did so and we have critically reviewed both sections 4 and 5 removign the parts that fit less the scope of the review.
We have also contemplated the suggestion by the reviewer to expand the scope. However, given the confusion on the boundaries of the scope, raised as well by the other reviewer, combined with the already sizable length of the manuscript, we have decided against this. Instead, we have attempted to better define and narrow the scope.
- Table 2, “DNA self assemblies” was placed in the category of “Not yet used in microrobotics”. But ref 156 shows that DNA was self-assembled into microswimmers.
R: We thank the reviewer for this observation. Even though we agree that ref 156 is indeed an application of DNA self-assemblies as microswimmers, we do not consider the technique mature enough to be compared to the methods in the other category. We acknowledge that our phrasing is ambiguous and therefore change “used” into “established”.
Round 2
Reviewer 1 Report
Comments and Suggestions for Authors
The authors have well addressed the issues. The paper is now recommended for publication.
Reviewer 2 Report
Comments and Suggestions for Authors
The authors addressed the issues I mentioned. I think it is good enough for publication.